# Anthracyclines induce global changes in cardiomyocyte chromatin accessibility that overlap with cardiovascular disease loci

E. Renee Matthews[1], Raodatullah O. Abodunrin[2], John D. Hurley[3], Sayan Paul[1], José A. Gutiérrez[1], Alyssa R. Bogar[1], Michelle C. Ward[1]*

**1** Department of Biochemistry and Molecular Biology, University of Texas Medical Branch, Galveston, Texas, United States of America, **2** Post-Baccalaureate Research Education Program, University of Texas Medical Branch, Galveston, Texas, United States of America, **3** Biochemistry, Cellular and Molecular Biology Graduate Program, University of Texas Medical Branch, Galveston, Texas, United States of America

* miward@utmb.edu

## Abstract

Breast cancer drugs including anthracyclines (ACs) and Trastuzumab increase the risk for cardiovascular diseases (CVDs) such as atrial fibrillation (AF) and heart failure (HF) that ultimately affect the heart muscle. These CVDs are associated with hundreds of genetic variants in non-coding regions of the genome. However, how these drugs affect the regulatory potential of the non-coding genome of the heart and CVD risk loci is unknown. We therefore measured global chromatin accessibility across iPSC-derived cardiomyocytes derived from four healthy individuals that we treated with topoisomerase II (TOP2) inhibiting ACs, Doxorubicin, Epirubicin, and Daunorubicin, and the anthracenedione, Mitoxantrone as well as the TOP2-independent monoclonal antibody Trastuzumab, for three and 24 hours. We identified tens of thousands of open chromatin regions that are differentially accessible in response to TOP2 inhibitor treatments over time, and no changes in response to Trastuzumab. Early AC-responsive regions are promoter-proximal and are enriched for regions bound by TOP2 beta. Late AC-responsive regions are enriched for AC response genes particularly those associated with mismatch repair. AC-response regions near AC response genes are enriched for FOS:JUNB transcription factor motifs. Three AC-induced cardiotoxicity-, 28 AF- and 14 HF-associated SNPs directly overlap late AC-responsive regions. Early AC-responsive regions are enriched for AF SNPs including rs3176326, which is also associated with HF, and is an eQTL for *CDKN1A* in heart tissue. This SNP associates with increased chromatin accessibility at a TOP2 beta-bound region, increased histone acetylation, and increased *CDKN1A* expression in response to all ACs. Our results demonstrate large-scale changes in chromatin accessibility in cardiomyocytes treated with ACs, which correspond to several regions harboring AF and HF risk loci. The identified

**Data availability statement:** All ATAC-seq data have been deposited in the Gene Expression Omnibus (www.ncbi.nlm.nih.gov/geo/) under accession number GSE291260, H3K27ac CUT&Tag data under accession number GSE291262, and TOP2B ChIP-seq data under accession number GSE303207. All custom analysis scripts used for this project are available at https://github.com/mward-lab/Matthews_cardiotox_ATAC_2025 made possible by the workflowr package.

**Funding:** This work was funded by a Cancer Prevention and Research Institute of Texas (CPRIT) Recruitment of First-Time Faculty Award (RR190110) to M.C.W and the National Institutes of Health grant R35GM150459 to M.C.W. J.D.H was supported by a Jeane B. Kempner Predoctoral Fellowship administered through UTMB. The funders had no role in study design, data collection and analysis, decision to publish, or preparation of the manuscript.

drug-responsive chromatin regions can be used to annotate variants in cancer patient populations to contribute to risk estimation for CVD.

## Author summary

Anthracyclines are a widely used class of breast cancer drugs that are linked to cardiac toxicity and the development of heart disease in some women. There are hundreds of genetic variants that associate with risk for heart disease; however their role and mechanism of action in drug-induced toxicity is unclear given that most reside in the non-coding genome. We therefore tested the effects of five breast cancer drugs on genome-wide chromatin accessibility using induced pluripotent stem cell-derived cardiomyocytes. We find tens of thousands of chromatin regions that change in accessibility after anthracycline treatment and associate with changes in nearby gene expression. These genes function in DNA mismatch repair pathways in line with anthracyclines inducing DNA damage. We find 42 heart disease-associated genetic variants in regions that change in accessibility following all anthracycline treatments. This suggests that cancer drugs have large effects on the non-coding genome of heart cells including at regions associated with heart disease. This research contributes to our understanding of how genetic variants associated with disease exert their effects.

## Introduction

Anthracyclines (ACs) and the monoclonal antibody Trastuzumab (TRZ) are effective therapeutic agents used in the treatment of breast cancer. TRZ is widely used to treat HER2-positive breast cancers, while ACs including Doxorubicin (DOX) and its structural analogues Epirubicin (EPI) and Daunorubicin (DNR), are standard treatments for triple-negative breast cancers in addition to being used to treat other cancer types regardless of HER2 receptor status.

Treatment with either ACs or TRZ associates with increased risk for cardiovascular disease (CVD) and can ultimately lead to heart failure [1,2]. These effects may manifest following a single treatment or years after [3]. Epidemiological studies have indicated that breast cancer patients treated with ACs are more likely to develop heart failure than healthy individuals [4]. Even young breast cancer patients, without CVD risk factors, show decreased heart contraction following AC treatment [5]. In addition to heart failure, AC treatment also associates with increased risk for arrhythmias [6]. Notably the increased risk for CVD in breast cancer patients is associated with heart failure and arrhythmia, and not ischemic heart disease, indicating that damage to the myocardium itself is evident [7].

Cardiac dysfunction following cancer treatment can be referred to as cancer therapy-related cardiac dysfunction (CTRCD) [8]. While the exact definition varies across clinical guidelines, declining heart function, typically deduced through a

reduction of the heart's ejection fraction, is accepted as a key metric. To understand the genetic component of risk for breast cancer patients developing CTRCDs, Genome-wide association studies (GWAS) have been performed [9–11]. These studies have highlighted tens of genetic variants nominally-associated with AC toxicity [9,10]. Over a hundred genetic variants have also been robustly associated with heart failure and arrhythmia independent of cancer drug treatment, indicating a genetic component to CVD [12].

Deducing the mechanisms behind, and effects of, disease-associated genetic variants is challenging in human patients. Large numbers of individuals are required to overcome environmental effects, and controlled perturbation studies cannot typically be performed for ethical and technical reasons. The ability to generate disease-relevant cell types from induced pluripotent stem cells has facilitated progress in this area. Cellular models using induced pluripotent stem cell-derived cardiomyocytes (iPSC-CMs) have recapitulated the clinically-observed AC- and TRZ-induced cardiotoxicity phenotype, validating this *in vitro* approach [13,14]. These cellular models have allowed for the identification of genetic variants that associate with gene expression in response to DOX providing further support for the role of genetic variation and its molecular effects on cardiotoxicity [15]. We have shown that treatment with ACs (DOX, DNR, EPI) impacts the transcriptome in a similar manner across drugs and affects cardiomyocyte function [16]. However, the molecular basis for the gene expression changes, and how they contribute to cardiotoxicity is unclear.

ACs have classically been characterized as DNA-damaging agents that induce DNA double strand breaks through their interactions with topoisomerase II (TOP2) [17,18]. It is the beta isoform (TOP2B) that mediates the cardiotoxic effects on the heart [18]. However, there is more recent evidence that ACs induce their effects through damage to both DNA and chromatin, and that decoupling these effects can limit cardiotoxicity [19]. Survival of breast cancer patients after DOX treatment has been shown to be mediated through chromatin regulators [20]. Chromatin regulators are also enriched amongst genes that respond to ACs in iPSC-CMs implicating a chromatin mechanism in cardiomyocytes [16]. DOX has also been shown to affect chromatin accessibility around transcription factor motifs in breast cancer cells suggesting a mechanism for transcriptional changes [21]. However, the impact of ACs and TRZ on the chromatin landscape of cardiomyocytes is unknown.

We therefore designed a study to investigate the effects of five breast cancer drugs including three ACs, the anthracenedione Mitoxantrone (MTX) that also targets TOP2, and TRZ on the global chromatin landscape of iPSC-CMs in four healthy female individuals. We were able to identify thousands of genomic regions where chromatin accessibility changes in response to drug treatment, and investigate the chromatin context of CVD-associated genetic variants in the presence of cancer drug treatment.

## Results

### AC treatment induces thousands of chromatin accessibility changes in iPSC-CMs

To test the effects of breast cancer drugs on cardiomyocytes, we obtained iPSCs from four healthy female individuals and differentiated them into iPSC-CMs (Fig 1A and 1B). The median proportion of cells expressing cardiac troponin T across individuals is 98% indicating high-purity cultures (S1 Fig and S1 Table). Our previous work indicated that 24 hours of treatment of iPSC-CMs with a sub-lethal, clinically-relevant concentration (0.5 µM) of the TOP2 inhibitors (TOP2i) DOX, DNR, EPI and MTX adversely affects calcium handling features associated with cardiomyocyte contraction and induces thousands of gene expression changes, while TRZ does not [16]. Treatment with TOP2i for three hours induces only hundreds of gene expression changes. TOP2i exert their effects on cancer cells by trapping TOP2 onto DNA causing DNA double-strand breaks. We therefore compared the effects of TOP2i treatments on DNA damage in iPSC-CMs. We treated iPSC-CMs from each individual with 0.5 µM DOX, DNR, EPI, MTX, TRZ and a vehicle control (VEH) for three and 24 hours and measured the expression of the DNA damage marker gamma H2AX. DNA damage increases in all TOP2i-treated cells compared to VEH within three hours of treatment (t-test; $P < 0.05$; Fig 1C and 1D and S2 Table). DNA damage did not increase following TRZ treatment.

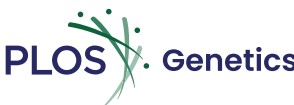

Fig 1. **Anthracycline treatment of iPSC-CMs results in DNA damage and chromatin changes.** (A) Experimental design of the study. iPSCs derived from four healthy women aged 20 to 30 were differentiated into cardiomyocytes (iPSC-CMs) and exposed to a panel of drugs used in breast cancer

treatment. Chromatin accessibility changes in response to TOP2 inhibitors (TOP2i) Doxorubicin (DOX), Epirubicin (EPI), Daunorubicin (DNR), and Mitoxantrone (MTX), a non-TOP2i, Trastuzumab (TRZ), and a water vehicle (VEH) were measured by ATAC-seq. (B) Expression of cardiac-specific markers cardiac troponin T (green) and alpha-actinin (red) in iPSC-CMs from each individual. Nuclei are stained with Hoechst (blue). Scale bar: 20 µm. (C) Expression of the DNA damage marker gamma H2A.X following treatment with 0.5 µM DOX, EPI, DNR, MTX, TRZ and VEH for three and 24 hours. Representative images from Individual A are shown. Arrows indicate gamma H2A.X positive nuclei. Scale bar: 20 µm. (D) Quantification of the proportion of DNA damage-associated nuclei following treatment with DOX, EPI, DNR, MTX, TRZ and VEH for three and 24 hours. Data representative of the mean proportion across at least 500 cells from each of four individuals. Asterisk represents drugs that induce an increase in DNA damage compared to VEH ($P < 0.05$). (E) Principal component analysis of ATAC-seq-derived accessibility measurements ($\log_2$ cpm) across 48 samples representing four individuals (A, B, C, D), the length of exposure (three hours: circles, 24 hours: triangles), and six treatments (DOX (mauve), EPI (pink), DNR (yellow), MTX (blue), TRZ (olive) and VEH (green)). Accessibility is based on 155,557 open chromatin regions across treatments.

To determine whether these DNA-damaging agents have effects on chromatin accessibility, we treated iPSC-CMs with 0.5 µM DOX, DNR, EPI, MTX, TRZ and VEH and collected cells for global chromatin analysis by ATAC-seq at three and 24 hours post-treatment in treatment-balanced batches across individuals (S1 Table). For each of the 48 samples, we obtained ATAC-seq reads that map uniquely to the nuclear genome, and designated open chromatin regions (See Methods; S2 Fig and S3 Table). The number of mapped fragments and open chromatin regions is similar across treatments within each timepoint (S3 Fig). The median number of open chromatin regions is higher in the samples treated for 24 hours than those treated for three hours (3 hour median = 73,241, 24 hour median = 88,737). In order to identify a high-confidence set of open chromatin regions we removed annotated blacklist regions that are known to exhibit non-specific signal, and selected those regions that are present in at least five of the 48 samples. This yielded a superset of 172,481 open chromatin regions. To assess the quality of the ATAC-seq data we determined the percentage of fragments in the set of open chromatin regions for each sample. All samples have at least 20% of fragments in open chromatin regions in line with standards of the field (S4 Fig).

In order to quantify changes in chromatin accessibility following treatment, we counted the number of reads in the superset of open chromatin regions across all samples. We filtered out regions with low read counts, yielding 155,557 regions for downstream analysis. To determine whether the accessible chromatin regions we identified are physiologically relevant, we compared our data to chromatin accessibility data from heart left ventricle tissue from a female donor [22]. We found that 43% of our accessible regions overlap with regions identified in heart tissue, which is greater than expected by chance (permutation test; $P < 0.05$; S5 Fig). These open chromatin regions are also enriched at transcription start sites (TSS; S6 Fig), including that of *TNNT2*, a cardiac-specific gene (S7 Fig). Unsupervised hierarchical clustering separates samples based on whether they are treated with TOP2i for 24 hours, and then by time and treatment group (S8 Fig). Principal component analysis revealed that the first principal component, representing 22.31% of variation in the data associates with treatment, and the second principal component, representing 13.53% of variation in the data, associates with individual and time (F-test; $P < 0.05$; Figs 1E and S9).

We next tested the effect of each drug on chromatin accessibility at each timepoint. We identified thousands of differentially accessible regions (DARs; adjusted $P < 0.05$) following three hours of TOP2i treatment (DOX vs VEH = 3,473; DNR vs VEH = 22,738; EPI vs VEH = 14,234; MTX vs VEH = 804; TRZ vs VEH = 1; Figs 2A and S10 and S4-S8 Tables), and tens of thousands after 24 hours of treatment (DOX vs VEH = 64,820; DNR vs VEH = 79,995; EPI vs VEH = 66,501; MTX vs VEH = 24,250; TRZ vs VEH = 0; S10 Fig and S9-S13 Tables). There was only one DAR following TRZ treatment at three hours and no DARs following TRZ treatment at 24 hours. We therefore further investigated the four sets of early DARs (DOX_3_DARs, DNR_3_DARs, EPI_3_DARs, MTX_3_DARs) and four sets of late DARs (DOX_24_DARs, DNR_24_DARs, EPI_24_DARs, MTX_24_DARs) together with the corresponding constitutively accessible regions (CARs) across TOP2i-treated iPSC-CMs. CARs are defined as all accessible regions that are not classified as DARs for each drug treatment.

Accessibility profiles in DARs show the expected clustering of VEH and drug-treated samples except for one EPI sample that clusters closer to VEH samples (S11 Fig). We considered all accessible regions for TRZ treatment given the

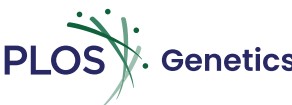

Fig 2. **Thousands of chromatin accessibility changes are induced in iPSC-CMs following anthracycline treatment.** (A) Proportion of the 155,557 open chromatin regions that are classified as a Differentially Accessible Region (DAR; adj. *P* < 0.05) in response to a drug treatment or a Constitutively

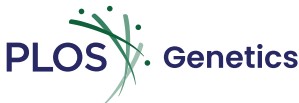

Accessible Region (CAR) that does not increase or decrease in accessibility in response to treatment. (B) Pearson correlation of log₂ fold change values across all treatment-time pairs with respect to the VEH treatment. Top color bars represent time (three hours: pink; 24 hours: brown), drug, and drug class (anthracycline: yellow; non-anthracycline: orange). (C) Comparison of DARs following three hours of treatment with each TOP2i drug. (D) Comparison of DARs following 24 hours of treatment with each TOP2i drug. (E) Comparison of DARs present in at least one drug at three hours and at least one drug at 24 hours. (F) Example of a DAR that is present following 24 hours of treatment with all TOP2i within an intron of the *ITPR1* gene. Asterisk represents treatments in which the region is designated as a DAR. Data are aggregated across four individuals within a treatment group.

lack of DARs following this treatment. Here, samples cluster by individual rather than treatment. The most significant DAR following each AC treatment is also a DAR in the other AC treatments and includes loci near the heat shock protein gene *DNAJB12* and the replication gene *FBXL12* (S12 Fig). Similarly, log₂ fold changes across ACs are highly correlated at each timepoint (Fig 2B) suggesting sharing of the chromatin response across ACs. Indeed, there are 2,665 DARs that are shared following three hours of AC treatment and 33,507 that are shared following 24 hours of AC treatment (Fig 2C and 2D). When considering the 25,831 DARs present in at least one three-hour drug treatment, 69% are also DARs following 24 hours of treatment suggesting persistence of chromatin changes (Fig 2E). However, there are 75,482 DARs unique to the 24 hour-timepoint; including a region in the *IPTR1* gene (Fig 2F).

## Early AC-responsive chromatin regions are enriched for TOP2B-bound regions near transcription start sites

ACs and MTX target the TOP2 protein. TOP2B is a highly expressed isoform of the protein in the heart that is associated with cardiotoxicity. TOP2B resolves topological stress by inducing DNA double-strand breaks and is also a transcriptional regulator [23]. Given the association between TOP2B and DNA, we asked whether DARs are at loci that are bound by TOP2B. We therefore determined the global binding profile of TOP2B in iPSC-CMs by ChIP-seq. We identified 5,410 binding sites genome-wide. Binding of TOP2B occurs at genes that are highly expressed in the heart, including *TNNT2* (Fig 3A). Genome-wide, 76% of TOP2B-bound regions localize to promoter regions (Fig 3B). When considering the overall distribution of DARs across the genome, three-hour AC DARs are more associated with promoter regions than 24-hour DARs (Fig 3B). We next integrated the TOP2B-bound regions with the set of accessible chromatin regions. The majority of TOP2B-bound regions are localized within accessible regions (n = 5,134; 95%; Fig 3C). We therefore tested whether TOP2B-bound regions are enriched in the four TOP2i-associated DAR sets compared to the four CAR sets at each timepoint. We find that all three-hour DAR sets are enriched for TOP2B-bound regions (Fisher's exact test; $P < 0.05$; Fig 3D). However, only MTX DARs are enriched for TOP2B-bound regions at 24 hours, and all AC DARs are depleted ($P < 0.05$). RNA-seq data from a matched differentiation and treatment experiment indicates that *TOP2B* mRNA levels are not affected by TOP2i treatment at three hours but are decreased in response to all TOP2i at 24 hours; however ACs are associated with a larger effect size than MTX (median log₂ fold change AC = -0.94, log₂ fold change MTX = -0.39) [16], suggesting that TOP2B levels may explain the discrepancy between AC and MTX response following 24 hours of treatment (Fig 3E). These data suggest that early changes in accessibility in response to TOP2 inhibition associate with regions that are bound by TOP2B, while many accessibility changes occur independent of TOP2B over time.

We therefore next sought to identify the sequence features associated with regions that change in their accessibility across time (S14 Table). Given that approximately half of the human genome comprises transposable elements (TEs), we first calculated TE enrichment in each set of DARs relative to the corresponding CARs. All DARs except for three-hour DOX DARs are enriched for all TEs (Chi-square test; adjusted $P < 0.05$; Fig 3F). A TE is said to overlap a chromatin region when it overlaps by 1 bp. Results are similar when requiring a chromatin region to overlap 50% of the total length of a TE. We next considered each class of TE independently. Only two TE classes are enriched following three hours of treatment. SINEs are enriched in response to all ACs but not MTX, and SVAs are enriched in response to DNR. LTRs are enriched in response to all ACs but not MTX at 24 hours. We further investigated the constituent families across TE classes. MIRs are

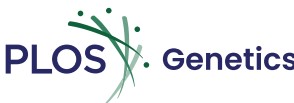

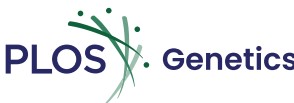

Fig 3. **Early anthracycline-responsive chromatin regions are enriched in TOP2B-bound regions near transcription start sites.** (A) TOP2B binding profile and Input DNA profile at the *TNNT2* gene following ChIP-seq in iPSC-CMs. Regions of enrichment are denoted as TOP2B-bound regions.

All accessible chromatin regions from ATAC-seq experiments together with the accessibility profile in VEH-treated iPSC-CMs at three and 24 hours are also shown. (B) Distribution of gene features across TOP2B-bound regions and DARs. 'Downstream' refers to regions where the center of the region is located less than 300 bp from the transcription termination site. 'Distal intergenic' refers to regions not localized within any other gene feature. (C) Overlap of accessible chromatin regions and TOP2B-bound regions. Regions in one set may overlap one or more regions in the other set. (D) Enrichment of TOP2B-bound regions in each set of DARs compared to CARs for each treatment. Asterisk represents a significant enrichment or depletion (Fisher's exact test; \*\*\*$P < 0.001$). (E) mRNA expression levels of *TOP2B* in response to drug treatments in a previously reported matched experiment [16]. Asterisk represents treatments where *TOP2B* is differentially expressed compared to VEH (adj. $P < 0.05$). (F) Enrichment of genomic features in DARs compared to CARs for each drug treatment determined by Chi-square test and Benjamini-Hochberg multiple testing correction. Asterisk represents significant enrichment in DARs relative to CARs (adj. $P < 0.05$). Transposable elements (TE) are represented in aggregate, and stratified into five TE classes; short-interspersed nuclear elements (SINEs), long-interspersed nuclear elements (LINEs), DNA transposons (DNAs), long terminal repeats (LTRs), and SINE-VNTR-*Alu* retrotransposons (SVAs). The 'Other features category' consists of CpG islands (CGI) and transcription start sites (TSS). Human heart candidate *cis*-regulatory elements (cREs) from a female adult were obtained from the ENCODE SCREEN database [22]. cREs are represented in aggregate (All cREs) and stratified into promoter-like sequences (PLS), distal enhancer-like sequences (dELS), proximal enhancer-like sequence (pELS), and CCCTC-binding factor sequences (CTCF).

enriched in three-hour AC DARs compared to all accessible regions, while Alu, L2 and ERVL-MaLR families are enriched across 24-hour DARs. (S13 Fig). Several of these TE families, including MIRs, can act as gene regulatory sequences.

We therefore investigated additional sequence features associated with gene regulation. We find that CpG islands are enriched across most three-hour DAR sets and the 24-hour MTX DAR set but it is notably not enriched across any 24-hour AC DAR sets (Fig 3F). As most CpG islands are located close to transcription start sites (TSS), we tested for enrichment of this feature across DAR sets. All DARs enriched for CpG islands are also enriched for TSS (Fig 3F). We next asked how these chromatin regions correspond to regulatory elements previously identified from DNase I hypersensitivity data and ChIP-seq data for H3K27ac, H3K4me3 and CTCF in human heart tissue from the ENCODE SCREEN database (See Methods). We considered all defined *cis*-regulatory elements (cREs) and found all three-hour DARs and 24-hour MTX DARs to be enriched but not 24-hour AC DARs (Fig 3F). Proximal enhancer-like sequences (pELS), and distal enhancer-like sequences (dELS) are enriched across these DAR sets. dELS are the only cREs enriched across 24-hour AC DARs (Fig 3F). Together, these results suggest that the set of chromatin regions that change in their accessibility early in response to drug treatment has distinct sequence features and genomic distribution compared to regions that change later.

## AC-induced chromatin accessibility changes correspond to changes in active histone modifications and gene expression

Chromatin regions that change in accessibility in response to drugs associate with regulatory regions identified in heart tissue. However, novel regulatory regions may be unmasked in response to drugs. We therefore selected a subset of individuals and profiled the enrichment of the active histone modification H3K27ac by CUT&Tag in TOP2i- and VEH-treated iPSC-CMs (Fig 4A). We used cells from the same differentiation and treatment batch used to generate the ATAC-seq data (S15 Table). We identified regions enriched for H3K27ac as we did for chromatin accessibility (See Methods; S16 Table). This yielded 20,137 regions enriched for H3K27ac in at least five samples. These regions are enriched around TSS as expected (S14 Fig). 98.8% of H3K27ac regions overlap with our regions of accessible chromatin. Following removal of outlier samples, the samples primarily separate based on whether they were treated with TOP2i at 24 hours, similar to the chromatin accessibility data (S15 Fig). We identified hundreds of differentially enriched regions (DERs; adjusted $P < 0.05$) following three hours of TOP2i treatment (DOX vs VEH = 114; DNR vs VEH = 1,277; EPI vs VEH = 0; MTX vs VEH = 0; S16A Fig and S16-S19 Tables), and hundreds to thousands after 24 hours of treatment (DOX vs VEH = 840; DNR vs VEH = 3,820; EPI vs VEH = 383; MTX vs VEH = 1; S20-S23 Tables). In line with the chromatin accessibility data, the magnitude of the response is highly correlated between ACs within each timepoint (S16B Fig).

To determine whether the AC-responsive chromatin accessibility regions comprise regulatory regions, we overlapped our set of accessible chromatin regions (155,557) and histone-enriched regions (20,137) and compared the response to each drug. Drug responses separate by treatment time, regardless of whether chromatin accessibility or active chromatin

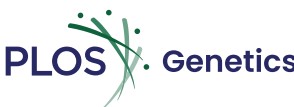

Fig 4. Anthracycline-induced chromatin accessibility changes correspond to changes in active histone modification enrichment and gene expression. (A) Overview of the experimental set-up to collect H3K27ac CUT&Tag data and integrate with chromatin accessibility and gene expression data [16]. As for ATAC-seq, iPSC-CMs were treated for three or 24 hours with 0.5 μM DOX, EPI, DNR, MTX, or VEH. H3K27ac-enriched regions are overlapped with open chromatin regions and connected to iPSC-CM expressed genes whose TSS is located within 2 kb of the region. (B) Correlation

between the median log$_2$ fold change (log$_2$ FC) across drugs for shared region-gene pairs for chromatin accessibility and H3K27ac enrichment (left) and chromatin accessibility and gene expression (right) at each timepoint. (C) Enrichment of 24 hour DARs near genes classified as differentially expressed in response to drug treatment at 24 hours compared to CARs (Fisher's exact test; **$P$<0.01, ***$P$<0.001) [16]. (D) Enrichment of transcription factor motifs in DARs near 24-hour differentially expressed genes (DEGs). Enriched motifs (– log$_{10}$(E-value) < 0.05) with an enrichment ratio value > 1.25 compared to DARs not near DEGs were selected. The consensus motifs unique to each motif cluster based on E-value are shown for each DAR set. Top x-axis displays percentage of DARs that have the consensus motif (data represented as circles). Bottom x-axis displays enrichment (– log$_{10}$(E-value)) as calculated by XSTREME (data represented as bars). (E) Biological pathways enriched amongst DAR-associated DEGs across drugs compared to all expressed genes (Benjamini-Hochberg q value < 0.05). Asterisk represents the DEG sets in which the pathways are enriched. Only the top 5 DNR-enriched pathways are shown.

is measured (S17 Fig). This suggests that a subset of AC-responsive chromatin changes coincide with gene regulatory regions.

Considering the changes in chromatin accessibility and active histone mark enrichment in response to drug treatment, we asked whether these changes coincide with gene expression changes. To do so, we linked all accessible chromatin regions with the closest TSS within 2 kb, and correlated the response effect size (median log$_2$ fold change across all TOP2i drugs) at each timepoint. The median drug response effect is highly correlated between chromatin accessibility and H3K27ac enrichment (3 hour, rho = 0.44; $P$<0.001 and 24 hour, rho = 0.58; $P$<0.001; Fig 4B), while the correlation between chromatin accessibility drug response and gene expression response is much stronger at 24 hours (3 hour, rho = -0.05; $P$<0.01 and 24 hour, rho = 0.26; $P$<0.001; Fig 4B). Open chromatin regions that overlap H3K27ac show a stronger correlation with gene expression (rho = 0.28 at 24 hours; $P$<0.001) than regions that do not overlap H3K27ac (rho = 0.14; $P$<0.001; S18 Fig). Combining drug responses across phenotypes for chromatin accessibility, active chromatin, and gene expression reveals a time-dependent effect that is shared across TOP2i (S19 Fig).

## AC DARs are enriched amongst differentially expressed genes

We next asked about the impact of DARs on gene expression. We obtained a set of genes that are differentially expressed in response to each treatment [16] and tested whether DARs are enriched close to the TSS of these genes compared to CARs. When considering either a two or 20 kb window around the TSS, DARs are enriched for differentially expressed genes following all TOP2i treatments at 24 hours compared to CARs (Fisher's exact test; $P$<0.05; Fig 4C). However, only DNR DARs are enriched for differentially expressed genes following three hours of treatment, at either window ($P$<0.05; Fig 4C).

To identify the transcriptional regulators that may lead to these gene expression changes at 24 hours, we searched for transcription factor motifs that are enriched amongst DARs that are associated with differentially expressed genes within 20 kb compared to DARs that do not associate with differentially expressed genes. The most enriched motif across all ACs is the FOS:JUNB motif (Fig 4D). This motif is not amongst the enriched motifs in MTX. Motifs for p53/p63 family members are enriched across all ACs but again these motifs are not amongst MTX-enriched motifs suggesting differences in the gene regulatory effects of ACs and anthracenediones. The most enriched motifs in MTX treatment associate with KLF15, NFYA and SP1 transcription factors (Fig 4D). KLF15 is amongst the top five most enriched transcription factor motifs in TOP2B-bound regions and associates with the most regions (12%), suggesting TOP2B-mediated effects for MTX at 24 hours.

To determine the impact of chromatin-accessibility mediated gene expression changes, we asked which pathways are associated with differentially expressed genes. We found that genes associated with the DNA mismatch repair pathway are enriched across ACs but not MTX (adjusted $P$<0.05; Fig 4E). DNR associated with additional pathways including glycerophospholipid metabolism and Rap1 signaling (Fig 4E). MTX-responsive regions associate with p53 signaling, DNA replication and cell cycle pathways.

## CVD-associated SNPs localize to AC-responsive chromatin regions

We next integrated our chromatin accessibility data with a GWAS that associated AC use with CTRCD. This study reported that there are 108 SNPs associated with AC-induced congestive heart failure [10]. We asked whether any of

these SNPs are associated with drug-responsive chromatin regions. Only four SNPs directly overlap our accessible chromatin regions (Fig 5A). Three of these SNPs overlap a DAR in at least one treatment. This includes rs12051934, located within the *NEDD4L* gene, that overlaps a DAR that increases in response to all ACs at 24 hours. Two SNPs overlap DARs in a subset of treatments: rs7777356, in the *CALD1* gene that is in an EPI_3_DAR, EPI_24_DAR and DNR_24_DAR, and rs7291763 in the *CDPF1* gene, that is a DNR_24_DAR.

We reasoned that SNPs linked to the reported cardiotoxicity-associated SNP in the same genomic region may also be relevant given that many disease-associated SNPs are in linkage disequilibrium with SNPs in open chromatin regions [26] and that linkage disequilibrium between SNPs decays with genetic distance. We therefore next considered all accessible regions within the same topologically associated domain (TAD) as the SNP. We obtained a set of TADs from left ventricle heart tissue [24,25] and calculated the distance between the SNPs and the sets of DARs and CARs in each TAD. We found that, within a TAD, three-hour AC DARs are closer to cardiotoxicity-associated SNPs than AC CARs, and that there is no difference in the distance between three-hour MTX DARs and corresponding CARs (Fig 5B). There are 14 regions that are DARs in response to all ACs in SNP-containing TADs (S20 Fig). One TAD contains three cardiotoxicity-associated SNPs rs4916358, rs10753081 and rs10798282 near the *PRDX6* gene. Chromatin accessibility increases in response to all ACs in an accessible region within 795–18,937 bp of each of the three SNPs (Fig 5C and 5D). All of these SNPs are expression quantitative trait loci (eQTLs) for the *DARS2* gene in heart left ventricle and atrial appendage. We find that concomitant with the increase in accessibility around these SNPs in response to AC treatment at 24 hours, there is a decrease in expression of the *DARS2* gene (Fig 5E). The TSS of the *DARS2* gene overlaps a chromatin region that is classified as a DAR in response to DOX, DNR and MTX at 24 hours and decreases in accessibility (Fig 5F). This chromatin region overlaps a TOP2B-bound region providing further support for the association of this gene with cardiotoxicity. Loss of *DARS2* expression leads to deregulation of mitochondrial protein synthesis [27]. The closest gene to these three SNPs is *PRDX6* (*DARS2* is over 350 kb away), which is not an eQTL in heart tissue but is an eQTL in seven other tissues. Its expression is increased in response to all AC treatments at 24 hours.

CTRCD has a range of clinical definitions, therefore we considered two types of CVD that individuals receiving ACs are at increased risk for: heart failure (HF) and atrial fibrillation (AF) as well as two unrelated CVD: ischemic heart disease (IHD) and coronary artery disease (CAD). We identified the set of SNPs associated with each disease from the GWAS catalog (403 SNPs for HF, 679 SNPs for AF, 2,320 SNPs for CAD and seven SNPs for IHD). We next integrated these SNPs with our chromatin accessibility data considering only SNPs that directly overlap accessible chromatin regions. We asked whether any CVD GWAS SNP sets were enriched amongst DARs compared to CARs. AF-associated SNPs are enriched amongst three-hour AC DARs but not MTX (Chi-square test; $P<0.05$; Fig 6A). MTX DARs at 24 hours are enriched for AF and HF. There are no DARs that are enriched for IHD or CAD SNPs. The 76 accessible regions overlapping AF SNPs show increased absolute effect sizes in response to ACs and MTX compared to TRZ at three and 24 hours (Wilcoxon-rank sum test; $P<0.05$; Fig 6B). Overall, there are 28 AF SNPs that overlap DARs in all ACs across timepoints (S21 Fig), and 14 HF SNPs that overlap AC DARs (S22 Fig).

One such SNP, rs3176326, is associated with AF and HF and overlaps an AC DAR following three and 24 hours of treatment. This DAR increases in accessibility in response to all AC treatments (Fig 6C and 6D) and increases in H3K27ac enrichment (Fig 6E). This SNP is an eQTL for *CDKN1A* in heart left ventricle. We find that as the accessibility of this region increases, the expression of *CDKN1A* increases (Fig 6F). This SNP is adjacent to a TOP2B-bound region (Fig 6C). These results indicate that genetic variants associated with CTRCD and CVD overlap regions of chromatin that change in their accessibility in response to drug treatment suggesting a gene regulatory role.

## Discussion

Individuals treated with ACs including DOX are at increased risk for developing CVD including AF and HF. DOX is a TOP2 poison, which leads to DNA double-strand breaks. The cardiotoxicity associated with DOX is mediated through



**Fig 5. Anthracycline-responsive regions are near anthracycline-induced cardiotoxicity-associated SNPs.** (A) 108 AC-induced cardiotoxicity-associated SNPs were obtained [10]. The four SNPs that overlap accessible regions are shown together with the response to treatment with each drug (log$_2$ fold change). Asterisk represents regions that are classified as DARs. (B) SNPs were associated with all accessible regions within a TAD identified in heart left ventricle tissue [24,25]. Accessible regions were stratified by whether they are designated as DARs or CARs in response to each treatment and the distance to the SNP measured. Asterisk represents treatments where the distance between sets differs (Wilcoxon-rank sum test; $P < 0.05$). (C) Example TAD containing three AC-induced cardiotoxicity-associated SNPs. Asterisk represents regions classified as DARs in response to at least one drug. Data aggregated across individuals. (D) Chromatin accessibility across drug treatments and time at the chromatin region within



10 kb of rs10753081. Asterisk represents treatments where the regions are denoted as DARs. (E) Expression (log₂ cpm) of the heart eGene *DARS2* (associated with rs4916358, rs10753081 and rs10798282) across drug treatments and time. Asterisk represents treatments where *DARS2* is denoted as a DEG. Expression data from Matthews *et al*. [16]. (F) Chromatin accessibility and TOP2B binding at the *DARS2* TSS. Asterisk represents treatments where the region is denoted as a DAR.

interactions between the drug and the TOP2B isoform present in the heart [18,28]. In addition to DNA damage, DOX also induces damage to chromatin [29], and enhances nucleosome turnover at promoters [30]. Chromatin regulators are also important contributors to survival of breast cancer patients treated with DOX [20]. We previously showed that treatment of iPSC-CMs with ACs induces thousands of global changes in gene expression over time, including genes encoding chromatin regulators [16]. In order to understand the contribution of chromatin to the clinical phenotype of AC-induced cardiotoxicity, and the AC-induced gene expression changes in iPSC-CMs, we profiled chromatin accessibility across five drugs used in the treatment of breast cancer. We identified and characterized tens of thousands of chromatin regions where accessibility is affected by AC treatment.

## TOP2i treatment affects the global chromatin accessibility landscape

We treated iPSC-CMs with clinically-relevant concentrations of four TOP2i and TRZ, and measured changes in accessibility at 155,557 chromatin regions three and 24 hours post-treatment. While less than a quarter of all chromatin regions show accessibility changes in response to any drug at three hours, nearly half of all chromatin regions show changes following 24 hours of treatment with at least one AC. 22% of DARs are shared across ACs at 24 hours. MTX induces half the number of chromatin accessibility changes compared to ACs; resulting in 12% of all DARs being shared across TOP2i. TRZ induces a change in chromatin accessibility at only one region following three hours of treatment and none following 24 hours of treatment.

The chromatin changes that we observe in response to ACs in iPSC-CMs are in line with the changes observed in cancer cells. DOX treatment of osteosarcoma cell lines induced ~20,000 chromatin accessibility changes with concomitant changes in chromatin organization and condensation [31]. The authors of this study showed that etoposide, an unrelated TOP2i, had minimal effects on chromatin accessibility. We find that MTX, another unrelated TOP2i, has fewer effects on chromatin accessibility compared to ACs. Thousands of chromatin accessibility changes also exist between breast cancer cell lines that are sensitive to DOX treatment compared to those that are resistant thereby providing further support for DOX effects on chromatin [21]. Interestingly, hypoxia and oxidative stress in iPSC-CMs induce minimal effects on chromatin accessibility across 15 individuals despite thousands of gene expression changes, which suggests that stress does not uniformly affect chromatin accessibility in cardiomyocytes [32].

TOP2i affect the TOP2B protein in the heart. We therefore profiled the global distribution of TOP2B binding in cardiomyocytes. We found that 95% of TOP2B-bound regions overlap open chromatin regions and that AC and MTX DARs at three hours are enriched for TOP2B-bound sites, while 24-hour AC DARs are depleted. This suggests that early changes in chromatin accessibility are directly linked to TOP2 inhibition. Early drug-responsive regions associate with elements associated with gene regulation including TSS, CpG islands and promoter and enhancer elements in the heart. These features are also enriched following 24 hours of MTX treatment but not AC treatment suggesting divergence in the response to these drugs over time.

## TOP2i-induced chromatin changes associate with gene expression changes

We integrated our drug-induced chromatin regions with drug-induced gene expression changes measured in the same experiment [16]. There are minimal gene expression changes following three hours of TOP2i treatment but thousands of changes following 24 hours of treatment. We found that AC and MTX DARs are enriched near genes that are classified as



**Fig 6. Anthracycline-responsive regions overlap SNPs associated with atrial fibrillation.** (A) Atrial fibrillation (AF; light blue; gold), ischemic heart disease (IHD; green), coronary artery disease (CAD; red) associated SNPs were obtained from the GWAS catalog [12]. Enrichment of CVD SNPs in DARs were compared to CARs for each treatment (Fisher's exact test; *$P < 0.05$, **$P < 0.01$). (B) Absolute effect size of chromatin accessibility changes in response to drug treatment in 76 regions overlapping AF SNPs. Asterisk represents a significant difference in the effect size for each TOP2i treatment compared to TRZ treatment (Wilcoxon-rank sum test; ***$P < 0.001$). (C) Chromatin accessibility at rs3176326 that is associated with both AF and HF within the *CDKN1A* gene in response to drug treatment. Data aggregated across individuals. Asterisk represents the treatments in which the region is denoted as a DAR. (D) Chromatin accessibility at the rs3176326 SNP across drug treatments. Asterisk represents treatments where

the region is considered a DAR. (E) H3K27ac enrichment (log$_2$ cpm) across drug treatments and time at the chromatin region that overlaps rs3176326. Asterisk represents the treatments in which the region is denoted as a differentially enriched region. (F) Expression (log$_2$ cpm) of the heart eQTL gene *CDKN1A* (associated with rs3176326 and rs730506) across drug treatments and time. Asterisk represents the treatments in which the gene is denoted as a DEG. Expression data from Matthews *et al*. [16].

differentially expressed in response to drug treatment suggesting that these regions have consequences on downstream molecular phenotypes. Pathway enrichment analysis indicated that chromatin-associated gene expression changes are enriched for pathways related to mismatch repair across ACs but not MTX. DNR is associated with several more pathways than the other ACs including glycerophospholipid metabolism likely due to the increased number of DARs following treatment with this drug. Gene expression-associated DARs in ACs are enriched for FOS and Jun AP-1 complex transcription factor motifs, while MTX DARs are not. FOS mediates DNA damage repair in other cell types [33], and JUNB is associated with DOX-induced p21 expression [34]. All AC DARs associated with gene expression changes are also enriched for p53 transcription factor motifs. P53 is known to be associated with the DNA damage response. Our results suggest that chromatin changes are associated with accessibility of the motif for this stress-responsive transcription factor. MTX associates with KLF15 binding, which is associated with heart failure [35]. Differences in response to ACs and MTX may relate to the fact that MTX also inhibits protein kinase C in addition to TOP2 [36]. TRZ has essentially no effect on chromatin accessibility in line with it not causing any gene expression changes [16].

## CVD-associated loci are within AC-responsive chromatin regions

Individuals treated with ACs are at increased risk for developing CVD including CTRCD, AF and HF. These CVDs have also been associated with hundreds of genetic loci in the genome. This suggests that there is both a genetic and environmental component of risk for developing CVD. Given that these loci are typically located in the non-coding genome, it can be challenging to determine the mechanistic basis for the association. Single-cell ATAC-seq experiments in human hearts have revealed that AF-associated SNPs are enriched in the cardiomyocyte population of cells suggesting that our *in vitro* system is appropriate to gain insight into disease risk [37,38]. We first asked whether loci associated with cardiotoxicity in breast cancer patients treated with DOX fall within AC-responsive chromatin regions [9,10]. We identified three variants that overlap AC-responsive chromatin regions directly. This includes rs12051934, that is within the *NEDD4L* E3 ubiquitin ligase gene. We found that in SNP-containing TAD chromatin regions, DARs are closer to these SNPs than CARs suggesting drug-responsive regulatory activity in these regions. This is highlighted in a locus near the *PRDX6* gene that shows chromatin accessibility changes in response to AC treatments close to cardiotoxicity-associated SNPs, and associates with the expression of the *DARS2* gene in heart tissue. The TSS of this gene is also associated with a DAR and overlaps a TOP2B-bound region. These results suggest that *DARS2* may be a cardiotoxicity-relevant gene despite not being the closest gene to the cardiotoxicity-associated SNPs.

We next investigated the overlap between our drug responsive chromatin regions and genetic loci associated with risk for AF and HF. There are hundreds of high-confidence variants associated with these traits. We identified 42 that overlap directly with AC-responsive chromatin. One such SNP is rs3176326 that is associated with risk for both AF and HF, and shows increased accessibility in response to TOP2i drugs. This SNP is also an eQTL for *CDKN1A*. *CDKN1A* expression increases in response to TOP2i and is a well-characterized p53 response gene [39]. These results suggest that genetic variability at these sites could affect chromatin accessibility and gene expression in response to drug treatment.

The role of chromatin modifications on risk for CVD has also been shown. H3K27ac ChIP-seq experiments in 36 failing and 34 non-failing hearts enabled the identification of histone quantitative trait loci (QTLs) [40]. 22 of these loci overlap AF GWAS SNPs implying a potential regulatory mechanism for the association. Genetic variants that associate with DOX-responsive gene expression (DOX response eQTLs) have been identified in iPSC-CMs treated with DOX [15].

We did not observe enrichment of DOX-response eQTLs compared to baseline eQTLs in our drug-responsive chromatin regions. This could suggest multiple gene regulatory pathways leading to DOX-responsive gene expression. Given the high degree of eQTL sharing across contexts, future work could investigate whether genetic variants associate with differences in chromatin accessibility (caQTLs). Indeed, cell-type-specific caQTLs mostly affect cell-type-specific open chromatin when comparing iPSC-CMs to the iPSCs and lymphoblastoid cells from which they were derived [41].

Personalized medicine promises to consider treatment options that best suit individual patients based on their individual genotypes and phenotypes. Genotyping of individuals to assess risk for disease or adverse drug events is becoming increasingly available. However, there is still a gap determining which SNPs are relevant to a phenotype. Our study identifies and provides thousands of genomic locations that robustly associate with treatment of a panel of ACs in cardiomyocytes. While this study alone is not able to predict CVD risk in specific women being treated for breast cancer, the data generated can contribute to functional annotation following genotyping of individuals in a cancer patient population to stratify risk, and inform on treatment schedules and early disease screening.

### Potential limitations of our study

We modeled AC-induced cardiotoxicity using iPSC-CMs. While we cultured our 27-day old cardiomyocytes in glucose-free media to more closely mimic the metabolic state of adult cardiomyocytes, they do not recapitulate all aspects of adult cardiomyocytes including binucleation and elongation. We collected ATAC-seq data from iPSC-CMs treated with TOP2i and TRZ for three and 24 hours to determine the drug-induced chromatin accessibility landscape. We chose an early and a late timepoint; however chromatin accessibility changes may be more dynamic, and hence some effects may be absent in the timepoints we chose. These timepoints do coincide with effects on DNA damage, gene expression and cellular function. While the chromatin accessibility measurements and gene expression measurements were from the same cellular differentiation, and were collected from cells treated at the same time, these were not from the same cells. More direct connections between chromatin accessibility and gene expression could be made from a single cell multi-omics study where both phenotypes are collected from the same cell. However, the strong concordance between chromatin accessibility and gene expression suggests a concerted cellular response to stress. We profiled chromatin accessibility and H3K27ac enrichment but there could be other effects on chromatin that are not captured using these approaches. Other histone modifications, transcription factors and DNA methylation may also contribute to drug-responsive gene regulation.

While many studies have demonstrated that DOX induces gene expression changes in cardiomyocytes that are relevant for CVD [13,15,16], the effects of ACs on chromatin, a genomic feature that can be directly associated with CVD SNPs, were unknown. We generated global chromatin accessibility data in iPSC-CMs from four individuals treated with four TOP2i drugs and TRZ, at two timepoints, to determine the impact of these drugs on chromatin. Our data supports the current understanding that ACs damage cardiomyocytes through TOP2B inhibition and the generation of DNA double-strand breaks. We uncovered strong effects of ACs on chromatin accessibility that associate with gene expression changes, and identified CVD-associated variants that localize in AC-responsive chromatin regions. These results suggest that ACs remodel chromatin in cardiomyocytes allowing for the direct functional annotation of genetic variants in individuals being treated with these drugs rather than indirect measurements of linked gene expression. We believe that this data from the iPSCORE panel of genotyped individuals will also provide a resource for determining the contribution of AC treatment to risk of developing CVD.

## Materials and methods

### Ethics statement

All cell lines used were generated by Dr. Kelly A. Frazer at the University of California San Diego as part of the National Heart, Lung and Blood Institute Next Generation Consortium [42]. The iPSC lines were generated with approval from the

Institutional Review Boards of the University of California, San Diego and The Salk Institute (Project no. 110776ZF) and informed written consent of participants. Cell lines are available through contacting Dr. Kelly A. Frazer at the University of California San Diego, or through the biorepository at WiCell Research Institute (Madison, WI, USA).

### Induced pluripotent stem cell lines

We used iPSCs from the iPSCORE resource that were derived from four unrelated, healthy female donors of Asian ethnicity between the ages of 21 and 32 years with no previous history of cardiac disease or breast cancer [42]. Individual A: UCSD143i-87–1 (iPSCORE_87_1, Asian-Chinese, age 21), Individual B: UCSD131i-77–1 (iPSCORE_77_1, Asian-Chinese, age 23), Individual C UCSD116i-71–1 (iPSCORE_71_1, Asian, age 32), and Individual D: UCSD129i-75–1 (iPSCORE_75_1, Asian-Irani, age 30).

### iPSC culture

Cells were maintained at 37 °C, 5% $CO_2$ and atmospheric $O_2$. iPSCs were maintained in feeder-free conditions using mTESR1 (85850, Stem Cell Technology, Vancouver, BC, Canada) with 1% Penicillin/Streptomycin (10–002-CI, Corning, Bedford, MA, USA) on Matrigel hESC-qualified Matrix (354277, Corning, Bedford, MA, USA) at a 1:100 dilution. Cells were passaged every 3–5 days using dissociation reagent (0.5 mM EDTA, 200 mM NaCl in PBS) when the culture was ~70% confluent.

### Differentiation of iPSCs into cardiomyocytes

Cardiomyocyte differentiation was performed as previously described [16]. Briefly on Day 0, as a 10 cm plate of iPSCs reached 80–95% confluence, media was changed to Cardiomyocyte Differentiation Media (CDM) [500 mL RPMI 1640 (15–040-CM Corning), 10 mL B-27 minus insulin (A1895601, ThermoFisher Scientific, Waltham, MA, USA), 5 mL Gluta-MAX (35050–061, ThermoFisher Scientific), and 5 mL of Penicillin/Streptomycin (100X) (30–002-CI, Corning)] containing 1:100 dilution of Matrigel and 12 μM CHIR 99021 trihydrochloride (4953, Tocris Bioscience, Bristol, UK). Twenty-four hours later (Day 1), the media was replaced with CDM. On Day 3, after 48 hours, spent media was replaced with fresh CDM containing 2 μM Wnt-C59 (5148, Tocris Bioscience). CDM was used to replace media on Days 5, 7, 10, and 12. Cardiomyocytes were purified through metabolic selection using Purification Media, a glucose-free, lactate-containing media [500 mL RPMI without glucose (11879, ThermoFisher Scientific), 106.5 mg L-Ascorbic acid 2-phosphate sesqui-magenesium (sc228390, Santa Cruz Biotechnology, Santa Cruz, CA, USA), 3.33 mL 75 mg/mL Human Recombinant Albumin (A0237, Sigma-Aldrich, St Louis, MO, USA), 2.5 mL 1 M lactate in 1 M HEPES (L (+) Lactic acid sodium (L7022, Sigma-Aldrich), and 5 mL Penicillin/Streptomycin] on Days 14, 16 and 18. On Day 20, purified cardiomyocytes were released from the culture plate using 0.05% trypsin/0.53 mM EDTA (MT25052 CI, Corning) and counted using a Countess 2 machine. Cardiomyocytes were plated in Cardiomyocyte Maintenance Media [CMM; 500 mL DMEM without glucose (A14430-01, ThermoFisher Scientific), 50 mL FBS (MT35015CV, Corning), 990 mg Galactose (G5388, Sigma-Aldrich), 5 mL 100 mM sodium pyruvate (11360–070, ThermoFisher Scientific), 2.5 mL 1 M HEPES (H3375, Sigma-Aldrich), 5 mL 100X GlutaMAX (35050–061, ThermoFisher Scientific), and 5 mL Penicillin/Streptomycin]. iPSC-CMs were matured in culture for a further 7–10 days, with CMM Media replaced on Days 23, 25, 27, and 29.

### Differentiation efficiency determination using cardiac troponin T expression

Cardiomyocyte purity was assessed using flow cytometry and cardiac troponin T staining using the protocol described previously [16]. Briefly, Day 25–27 iPSC-CMs were stained with a live stain (Zombie Violet Fixable Viability Kit (423113, BioLegend, San Diego, CA, USA)) and cardiac troponin T (TNNT2) antibody (Cardiac Troponin T, Mouse, PE, Clone: 13–11, BD Mouse Monoclonal Antibody, 564767, BD Biosciences, San Jose, CA, USA). Controls consisting of live/dead

stain-only, TNNT2 antibody only, and unlabeled cells were also included. Ten thousand cells were analyzed per sample on a BD LSR Fortessa Cell Analyzer. Purity is reported as the proportion of live cells that are positive for TNNT2. Values reported are the mean of two technical replicates for each individual.

### Immunofluorescence staining for cardiac troponin T and sarcomeric actinin expression

In a 24-well plate, 150,000 iPSC-CMs were seeded in CMM. Between Days 25–27, cells were fixed using 4% paraformaldehyde for 15 min and permeabilized with 0.25% DPBS-T for 10 min at room temperature. Cells were blocked with 5% BSA:DPBS-T for 30 min at room temperature, followed by incubation with a 1:400 dilution of Anti-Cardiac Troponin T rabbit polyclonal antibody (NC9860670, Fisher Scientific) and a 1:500 dilution of Monoclonal Anti-α-Actinin (Sarcomeric) antibody produced in mouse (A7811, Sigma-Aldrich) in 5% BSA:DPBS-T overnight at 4 ˚C. The next day cells were incubated with Donkey anti-Rabbit Alexa Fluor 594 (A21207, Thermo Fisher Scientific) and Donkey anti-Mouse Alexa Fluor 488 (A21202, Thermo Fisher Scientific) at a dilution of 1:500 each. Stained cells were imaged using the EVOS Cell Imaging System (Invitrogen) at 40x magnification.

### Drug stocks and usage

The panel of drugs used were Daunorubicin (30450, Sigma-Aldrich), Doxorubicin (D1515, Sigma-Aldrich), Epirubicin (E9406, Sigma-Aldrich), Mitoxantrone (M6545, Sigma-Aldrich) and Trastuzumab (HYP9907, MedChem Express, Monmouth Junction, NJ, USA). All drugs were dissolved in molecular biology grade water to a concentration of 10 mM per stock. DOX, DNR, EPI, and MTX stocks were stored at -80 ˚C with working stocks stored at 4 ˚C for up to one week. TRZ was stored at a 1 mM concentration at 4 ˚C for up to one month.

### Immunofluorescence staining for DNA damage marker gamma H2A.X

iPSC-CMs were plated at a density of 150,000 cells per well of a 24-well plate. At Day 25–27, cells were treated with 0.5 µM DOX, EPI, DNR, MTX, TRZ or VEH for three and 24 hours. After treatment, cells were fixed using 4% paraformaldehyde for 15 min and permeabilized with 0.25% DPBS-T for 10 min at room temperature. Cells were blocked with 5% BSA:DPBS-T for 30 min at room temperature, followed by incubation with a 1:500 dilution of Phospho-Histone H2A.X (Ser139) polyclonal rabbit antibody (2577, NC1602516, Cell Signaling-Fisher Scientific) in 5% BSA:DPBS-T overnight at 4 ˚C. The next day cells were incubated with Donkey anti-Rabbit Alexa Fluor 594 (A21207, Thermo Fisher Scientific) at a dilution of 1:1000 and Hoechst 33342 nucleic acid stain (P162249, Thermo Fisher Scientific). Stained cells were imaged using the EVOS Cell Imaging System (Invitrogen). Two fields of view at 10x, with a minimum of 500 cells total were taken across all treatment-time conditions. The total number of nuclei and γH2A.X positive cells were counted using a custom macros in image J. The number of γH2A.X positive nuclei were divided by the total number of nuclei to determine the percentage of γH2A.X positive nuclei across treatment-time. The mean between two fields of view for each treatment-time is reported by individual. Each condition was compared to VEH using a t-test, with $P < 0.05$ considered significant.

### Drug treatment of iPSC-CMs and cell collection

400,000 iPSC-CMs were plated per well of a 12-well plate on Day 20 following the initiation of differentiation. On Days 27–29, iPSC-CMs were treated with 0.5 µM DNR, DOX, EPI, MTX, TRZ, or VEH in fresh CMM. iPSC-CMs were collected three and 24 hours post-treatment, resulting in 48 samples from four individuals. iPSC-CMs were washed twice with ice-cold PBS and manually scraped using cold PBS on ice.

For the ATAC-seq samples, nuclei were extracted immediately as described below.

For the CUT&Tag samples, 100,000 manually-scraped iPSC-CMs were pelleted by centrifugation (500 g, 10 min). The supernatant was removed, and cells were resuspended in 50 µL cryopreservation-media (10% DMSO, 40% CMM, 50%

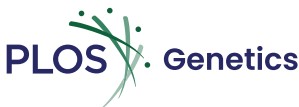

FBS). Cells were placed into a CoolCell (432000, Corning) at -80 ˚C for 24 hours, and then stored at -80 ˚C until further processing.

### ATAC-seq nuclei extraction and DNA tagmentation

We processed collected iPSC-CMs using the Active Motif ATAC-Seq Kit (53150, ATAC-Seq Kit, Active Motif, Carlsbad, CA, USA) following the manufacturer's directions. Briefly, approximately 100,000 scraped cells in PBS were pelleted by centrifugation at 500 g for 5 min. Cell pellets were resuspended in 100 µL cold ATAC Lysis Buffer. Cells were pelleted by centrifugation at 500 g for 5 min at 4 °C. Lysis buffer was removed and isolated nuclei were resuspended in 50 µL transposition mix. The tagmentation reaction was carried out at 37 °C for 30 min in a thermomixer set at 800 rpm. Transposed DNA was purified following the manufacturer's recommendation and resuspended in 35 µL elution buffer prior to storage at -20 °C.

### ATAC-seq library preparation

ATAC-seq library preparation across the 48 samples was performed in four treatment- and time- balanced batches corresponding to all drug treatments for each individual. Libraries were prepared following the Active Motif ATAC-Seq kit manufacturer's directions where DNA was amplified using a 14-cycle PCR reaction. Libraries were purified and bioanalyzed to determine library quality. Libraries were quantified by qPCR before being pooled and run on an Illumina Nextseq 2000 machine in four treatment- and time-balanced batches using 50 bp paired-end reads.

### TOP2B chromatin immunoprecipitation and sequencing

TOP2B chromatin immunoprecipitation (ChIP) was performed using 40 million iPSC-CMs from Individual A. iPSC-CMs were fixed with 1% formaldehyde for 10 min. 1 mL of 11% formaldehyde in Solution A (50 mM HEPES, 0.1 M NaCl, 1 mM EDTA, 0.5 mM EGTA) was added to 10 mL CMM in each of four 10 cm plates. Cells were quenched with 2.5 M glycine (500 µL per plate) and washed twice with ice-cold PBS. Cells were collected by scraping, pelleted following centrifugation at 2,500 rpm for 10 min at 4 °C, flash-frozen, and stored at -80 °C.

For immunoprecipitation, 100 µL Dynabeads Protein G (10004D, Thermo Fisher Scientific) were incubated overnight at 4 °C with 10 µg rabbit anti-TOP2B polyclonal antibody (ab72334, Abcam) in 0.5% BSA. Crosslinked cell pellets were sequentially lysed on ice using 1 mL Lysis Buffer 1 (50 mM HEPES-KOH pH 7.5, 140 mM NaCl, 1 mM EDTA, 10% glycerol, 0.5% NP-40, 0.25% Triton X-100), 1 mL Lysis Buffer 2 (10 mM Tris-HCl pH 8.0, 200 mM NaCl, 1 mM EDTA, 0.5 mM EGTA), and 200 µL Lysis Buffer 3 (10 mM Tris-HCl pH 8.0, 100 mM NaCl, 1 mM EDTA, 0.5 mM EGTA, 0.1% sodium deoxycholate, 0.5% N-lauroylsarcosine), each supplemented with freshly diluted cOmplete EDTA-free protease inhibitor cocktail (11873580001, Sigma-Aldrich). Chromatin was fragmented in Covaris microTUBEs (NC9871616, Thermo Fisher Scientific) using a Covaris S-series sonicator (10% duty cycle, intensity 5, 200 cycles/burst, 120 seconds total) to obtain average fragment sizes of 200–500 bp. Ten microliters of 10% Triton X-100 was added per 100 µL lysate, and 50 µL of each sample was reserved as an Input control. The remaining sonicated chromatin was incubated overnight at 4 °C with antibody-bound beads in a final volume of 1.5 mL Lysis Buffer 3 supplemented with 1% Triton X-100. Bead-bound complexes were washed six times in RIPA buffer (50 mM HEPES-KOH pH 7.5, 500 mM LiCl, 1 mM EDTA, 1% NP-40, 0.7% sodium deoxycholate) and once in TBS (20 mM Tris-HCl pH 7.6, 150 mM NaCl), then pelleted by centrifugation at 960 rpm for 3 min at 4 °C. Bound chromatin was eluted using 200 µL ChIP Elution Buffer (50 mM Tris-HCl pH 8.0, 10 mM EDTA, 1% SDS) for the ChIP sample and 150 µL for the Input sample. Samples were incubated at 65 °C for 15 min with intermittent mixing and then subjected to overnight crosslink reversal at 65 °C (6–18 hr). Following cross-link reversal, DNA was treated with 8 µL RNase A (1 mg/mL; AM2271, Thermo Fisher Scientific) at 37 °C for 30 min and 4 µL Proteinase K (20 mg/mL; AM2546, Thermo Fisher Scientific) at 55 °C for 1.5 hr. DNA was extracted using phenol:chloroform:isoamyl alcohol (15593031, Thermo Fisher Scientific) in Phase Lock Gel Light tubes (10847–800, VWR), and the aqueous phase was

precipitated with 2 volumes of 100% ethanol, 16 μL 5 M NaCl, and 1 μL GlycoBlue (AM9515, Thermo Fisher Scientific) at -80 °C for 30 min. DNA was pelleted at 20,000 rpm for 10 min at 4 °C, washed in 80% ethanol, and dried at ≤ 45 °C in a SpeedVac (SC210A-230, Thermo Fisher Scientific). DNA was resuspended in 50 μL Elution Buffer (19086, Qiagen) and stored at -20 °C for downstream library construction.

ChIP and Input DNA samples were processed for sequencing library preparation using the TruSeq ChIP Sample Preparation Kit (Illumina, IP-202–1012), following the manufacturer's protocol with minor adjustments. All ChIP DNA and 50 ng of Input DNA was used as starting material. End repair reactions were assembled by combining DNA with End Repair Mix and Resuspension Buffer, incubated at 30 °C for 30 min, and purified using a 1.6x volume of AMPure XP beads (A63881, Beckman Coulter). DNA was eluted in 18 μL Resuspension Buffer. Subsequent adenylation of 3' ends was performed by incubating the samples with A-Tailing Mix at 37 °C for 30 min, followed by 5 min at 70 °C. Indexed adapters: ACAGTG(A) for ChIP DNA and GCCAAT(A) for Input DNA were ligated in the presence of Ligation Mix, with reactions incubated at 30 °C for 10 min, and then terminated with Stop Ligation Buffer. Adapter-ligated DNA was purified in two consecutive rounds using 1:1 volume of AMPure XP beads and eluted in 23 μL of Resuspension Buffer. To enrich adapter-ligated fragments, 20 μL of purified product was amplified in a 50 μL reaction containing PCR Master Mix and PCR Primer Cocktail. Amplification was performed for 18 cycles using the following conditions: 98 °C for 30 s; 18 cycles of 98 °C for 10 s, 60 °C for 30 s, and 72 °C for 30 s; followed by a final extension at 72 °C for 5 min. Amplified libraries were cleaned with a 1:1 volume of AMPure XP beads and eluted in 30 μL Resuspension Buffer. Aliquots were reserved for downstream qPCR validation and gel-based size selection. For gel-based size selection, DNA was loaded onto 2% Ultra Agarose gels (1613107, Bio-Rad) prepared in 1x TAE buffer (1610743, Bio-Rad) containing SYBR Safe DNA stain (Invitrogen, S33102), and run alongside a Low Molecular Weight DNA ladder (N3233L, NEB) at 120 V for 40 min. DNA fragments ranging from 250 - 400 bp were visualized using a blue light transilluminator and excised using sterile scalpels. Excised gel slices were purified using the MinElute Gel Extraction Kit (28606, Qiagen), following the manufacturer's protocol. Elution was carried out using 20 μL of prewarmed Elution Buffer (50 °C). Final library yields were quantified using the Qubit dsDNA HS Assay Kit (Thermo Fisher Scientific), and fragment size distributions were assessed using the Agilent Bioanalyzer. The ChIP library was sequenced on the Illumina NextSeq 550 using 40 bp paired-end reads. The Input DNA library was sequenced on an Illumina MiniSeq using 40 bp paired-end reads. We obtained 40,380,049 reads for the ChIP sample and 20,968,544 reads for the Input sample.

## CUT&Tag library preparation

CUT&Tag libraries were generated for 30 samples in three treatment- and time-balanced batches across three individuals (TRZ-treated samples were not included for this assay). Benchtop CUT&Tag was adapted from the previously described protocol [43] and Kaya-Okur & Henikoff protocols.io (https://dx.doi.org/10.17504/protocols.io.bcuhiwt6).

Cryopreserved iPSC-CMs were fast-thawed and centrifuged at 4 °C for 4 min at 1000 g to remove cryopreservation media. The pellets were resuspended and washed twice in 1 mL of Wash Buffer [20 mM HEPES-KOH pH 7.5, 150 mM NaCl, 0.5 mM Spermidine, 1x Protease inhibitor cocktail; 11873580001, Roche]. Nuclei were extracted by incubating cells for 10 min on ice in 200 μL cold Nuclei Extraction buffer [NE Buffer; 20 mM HEPES-KOH pH 7.9, 10 mM KCl, 0.1% Triton X-100, 20% Glycerol, 0.5 mM Spermidine, 1x Protease Inhibitor cocktail]. Nuclei were pelleted by centrifugation at 4 °C for 4 min at 1,300 g. Nuclei were resuspended in PBS and fixed with 0.1% formaldehyde solution for 2 min at room temperature. Cross-linking was terminated by adding glycine to a final concentration of 0.1 M. After fixation, nuclei were centrifuged at 4 °C for 4 min at 1,300 g and resuspended in 1.5 mL of Wash Buffer.

Concanavalin A-coated magnetic beads (BP531, Bangs Laboratories Inc, Fishers, IN, USA) were prepared as previously described [44]. The fixed nuclei were suspended in 1.5 mL Wash Buffer and incubated with 10 μL of activated beads for 15 min at room temperature on an end-over-end rotator. Bead-associated nuclei were collected via magnet and unbound supernatant was discarded. The sample was resuspended in 50 μL of ice-cold Antibody Buffer [Wash Buffer with

2 mM EDTA and 0.1% BSA] with a 1:50 dilution of the H3K27ac antibody (39034, Lot # 31521015, Active Motif). Samples were incubated at 4 °C overnight on a tilt table. Excess primary antibody was removed following magnetic pelleting of bead-associated nuclei. Nuclei were resuspended in 100 µL of Guinea Pig anti-Rabbit IgG (Heavy & Light Chain) pre-absorbed secondary antibody (ABIN101961; Lot NE-200–032309, Antibodies-Online, Limerick, PA, USA) diluted 1:100 in Wash Buffer. The samples were incubated for 1 hr at room temperature on a tilt table. The secondary antibody was removed using a magnetic stand to clear the supernatant. The bound samples were washed twice for 5 min with 1 mL of Wash Buffer.

Tagmentation was started by incubating nuclei with 50 µL of a solution containing pAG-Tn5 (15–1017; Lot 23243004-C1, Epicypher, Durham, NC, USA) diluted 1:20 in '300 Buffer' [20 mM HEPES, pH 7.5, 300 mM NaCl, 0.5 mM Spermidine, 1x Protease inhibitor cocktail] on a tilt table for 1 hr at room temperature to bind with pAG-Tn5. Samples were collected using a magnetic stand and washed twice with 1 mL of '300 Buffer'. Samples were resuspended in 300 µL Tagmentation Buffer [10 mM MgCl$_2$ in 300 Buffer] to activate tagmentation and incubated in a water bath at 37 °C for 1 hr. To stop tagmentation, 0.5 M EDTA, 10% SDS, and 20 mg/mL Proteinase K was added sequentially to the sample for a final concentration of 0.01 M EDTA, 0.1% SDS, and 0.15 mg/mL Proteinase K. Samples were vortexed and incubated in a water bath at 55 °C for 1 hr. DNA was extracted from the aqueous layer following PCI-Chloroform separation. The samples were chilled on ice and centrifuged at 4 °C for 10 min at 16,000 g. DNA was resuspended in 100% ethanol and centrifuged at 4 °C for 1 min at 16,000 g. DNA pellets were retained and air dried before being dissolved in 30 µL TE buffer [1 mM Tris-HCl pH 8, 0.1 mM EDTA].

Libraries were generated by combining 21 µL of DNA from each sample with 2 µL of a 10 µM uniquely barcoded i5 primer and 2 µL of a uniquely barcoded i7 primer [45]. Each sample contained a unique barcode combination. A volume of 25 µL NEBNext HiFi 2 x PCR Master Mix (M0541S, New England Biolabs, Ipswich, MA, USA) was added and mixed. The sample was placed in a thermocycler with a heated lid using the following cycling conditions: 72 °C for 5 min; 98 °C for 30 s; 14 cycles of 98 °C for 10 s and 63 °C for 10 s; final extension at 72 °C for 1 min and hold at 8 °C. Post-PCR clean-up was performed by incubating libraries with 1.3 x volume of AMPureXP beads (A63881, Beckman Coulter, Brea, CA, USA) for 10 min at room temperature. Libraries were washed twice in 80% ethanol and eluted in 25 µL 10 mM Tris pH 8.0. To eliminate very large and small DNA fragments, libraries were processed through a double-sided cleanup by adding 0.55 x the volume of AMPureXP beads followed by 1.8 x the volume of beads.

Agilent Bioanalyzer High Sensitivity DNA Analysis was performed to assess library concentrations and size distributions. Five samples failed to make libraries (Individual_A_EPI_3hour; Individual_A_EPI_24hour; Individual_B_DOX_24hour; Individual_B_MTX_3hour; Individual_C_DOX_3hour) and were removed from further processing. Libraries (n = 25) were quantified by qPCR before being pooled together. The pooled libraries were sequenced with paired-end reads and a 75 bp read length on a single lane of the Illumina NextSeq550.

## ATAC-seq analysis

**Sequencing read processing and alignment.** Raw sequencing reads were assessed for quality using FastQC (https://www.bioinformatics.babraham.ac.uk/projects/fastqc/) and visualized with MultiQC [46]. Cutadapt [47] was run in PE legacy mode to remove any adapter sequences present. Paired-end sequencing reads were aligned to the hg38 genome using bowtie2 with the settings -D 20 -R 3 -N 1 -L 20 [48]. Reads mapping to mitochondria and more than one genome location were removed using samtools [49]. Duplicate reads were removed using Picard Tools (https://broadinstitute.github.io/picard/).

**Identification of accessible regions.** Regions of enrichment, i.e., accessible regions (peaks), were called on each file using MACS2 callpeak -f BAMPE -g hs –keep-dup all [50]. A master peak set was created with BEDtools [51] by first concatenating, then merging all peaks that were adjacent (bp difference between the two peaks being considered for merging sets is 0 bp) or overlapped by at least 1 bp. We selected a set of high-confidence peaks that are present in at

least five of 48 samples by first counting the total number of intersections between each narrowPeak file with the master peak set using BEDtools multiinter function. We then removed all regions from the intersection file which had a count of < 5. Next, BEDtools intersect was used between the master peak set and the filtered intersections count file using -wa -u flags to return only those regions from the master peak file that overlapped the filtered intersection file by at least 1 bp. We removed annotated blacklist regions (https://github.com/Boyle-Lab/Blacklist/) [52] to create a master set of high-confidence open chromatin regions (n = 172,481).

**Quantification of chromatin accessibility.** To quantify the accessibility of each chromatin region in each sample we counted the number of reads in the set of high-confidence open chromatin regions using Subread featureCounts [53].

**Fraction of reads in accessible chromatin regions.** The master set of high-confidence regions (n = 172,481) was used as features in Subread featureCounts [53] to generate the fraction of paired-end reads found in these high-confidence regions. featureCounts reports the total number paired-end reads, and the number of reads that overlap at least 1 bp of the provided feature file, for each bam file in the data set.

**Visualization of accessible chromatin loci.** The Integrative genomic viewer (IGV, version 2.19.1) [54,55] was used to visualize open chromatin regions across the genome for all samples using the bam and bigwig files.

**Filtering out open chromatin regions with low accessibility.** The high-confidence set of 172,481 open chromatin regions were filtered to exclude regions with mean $\log_2$ cpm values < 0 and three regions mapped to the Y chromosome. This gave us a set of filtered high-confidence open chromatin regions (n = 155,557) for downstream analysis.

**Integration with ENCODE heart ATAC-seq data.** We downloaded the ATAC-seq narrowPeak file (identifier ENCF966JZT) from ENCODE (www.https://www.encodeproject.org) experiment ENCSR204PZT (Michael Snyder Lab, Stanford, CA), which is derived from heart left ventricle tissue from a 41-year-old female. Using plyranges [56], we overlapped our 155,557 filtered high-confidence regions with the 218,982 regions in the narrowPeak file. Significance of the overlap was determined based on a permutation test from the regioneR package [57].

**TSS enrichment analysis.** TSS enrichment analysis of all genes was performed on each bam file using the TSSE function in R from the ATACseqQC package [58,59] using hg38-known genes from the annotation package TxDb. Hsapiens.UCSC.hg38.knownGene (R package version 3.20.0). These known gene regions were used for calculating the aggregate distribution of reads centered on a TSS location over the region that flanks the corresponding TSS (TSS score) according to https://www.encodeproject.org/data-standards/terms/#enrichment. TSS score = the depth of TSS (each 100 bp window within 1000 bp each side)/ the depth of end flanks (100 bp each end). TSSE score = max(mean(TSS score in each window)).

TSS enrichment of all genes within open chromatin regions was visualized using the plotAvgProf function from the ChIPseeker package in R [60,61].

**Identifying differentially accessible regions (DARs).** To identify differentially accessible regions we used an edgeR-voom-limma pipeline [62]. We first normalized the count data using TMM (Trimmed mean of M-values), then applied a voom transformation to calculate precision weights for linear modeling. Next, we modeled the individual as a random effect using the duplicateCorrelation function and transformed the data with the correlation adjustment. Finally, we contrasted each drug treatment against the vehicle at each timepoint. Differentially accessible regions (DARs) are defined as those regions for each treatment-vehicle pair that meet an adjusted $P$ value threshold of < 0.05.

## TOP2B ChIP-seq analysis

ChIP and Input DNA reads were first evaluated for quality using FastQC, followed by alignment to the GRCh38/hg38 human genome using Bowtie2 [48] in paired-end mode. The majority of reads mapped to the reference genome (ChIP: 96.8%; Input: 98.5%). To ensure high-confidence mapping, only uniquely aligned read pairs were retained using bam-tools filter [63]. Duplicate reads were removed using Picard MarkDuplicates with optical duplicates identified based on a 100-pixel distance threshold and scoring prioritized by base quality sums. Peak calling was performed using MACS2

in broad peak mode (--broad), with the Input DNA used as a control. Peaks were defined using a q-value cutoff of 0.05 and a broad region cutoff of 0.1. To estimate fragment size distributions, an mfold range of 5–50 and a bandwidth of 300 bp were applied. Motifs were discovered using MEME (Multiple Em for Motif Elicitation) and STREME (Sensitive, Thorough, Rapid, Enriched Motif Elicitation) and enriched motifs were identified using the SEA (Simple Enrichment Analysis) from the MEME Suite Software using primary peak sequences (BED format) with motif width constraints set to 8–15 bp.

### CUT&Tag sequencing analysis

**Sequencing read processing and alignment.** Raw sequencing reads for each of the 25 samples were assessed for quality using FastQC and visualized with MultiQC [46]. Cutadapt [47] was run in PE legacy mode to remove any adapter sequences present.

Paired-end sequencing reads were aligned to hg38 using bowtie2 with -q --threads 12 --very-sensitive-local --no-mixed --no-discordant --phred33 -I 10 -X 700 [48]. Reads were filtered using samtools [49] to remove reads mapping to more than one genome location.

**Identification of regions of H3K27ac enrichment.** Enriched regions were called on each file using MACS2 callpeak -f BAMPE -g hs –keep-dup all -q 0.05 [50]. A master enriched region set was created with BEDtools [51] by first concatenating, then merging all regions that were adjacent (0 bp difference) or overlapped by at least 1 bp. We selected a set of high-confidence acetylated-enriched regions that are present in at least five of 25 samples by first counting the total number of intersections between each narrowPeak file with the master enriched region set using BEDtools multiinter function. We then removed all intersections from the intersection file which had a count of < 5. BEDtools intersect was used between the master enriched region set and the filtered intersections file using -wa -u flags to return only those regions from the master enriched regions file that overlapped the filtered intersection file by at least 1 bp (n = 20,137).

**Quantification of H3K27ac enrichment.** To quantify the enrichment of H3K27ac in each sample we counted the number of reads in the set of high-confidence H3K27ac enriched regions using Subread featureCounts [53].

**Filtering out H3K27ac regions with low enrichment.** The 20,137 H3K27ac enriched regions all had mean $\log_2$ cpm values > 0 for each region across samples, and therefore no further filtering for low enrichment was performed.

Two samples were determined to be outliers and were removed (S15 Fig, Individual_B_VEH_24hours; Individual_C_VEH_3hours).

**Identifying differentially enriched H3K27ac regions.** We performed differential enrichment analysis on H3K27ac enriched regions for the following sets of treatment-time-individual sets given that a subset of libraries failed. For DNR-3 hour and DNR 24-hour, individuals A, B, and C were used. The remaining treatment-time-individual sets are as follows: DOX-3 hour, individuals A and B; DOX-24 hour, individuals A and C; EPI-3 hour and EPI-24 hour, individuals B and C; MTX-3 hour, individuals A and C; MTX-24 hour, individuals A and B; VEH-3 hour individual A and B; VEH 24 hour, individuals A and C. To identify differentially enriched H3K27ac regions, we used an edgeR-voom-limma pipeline [62]. We first normalized the count data using TMM (Trimmed mean of M-values), then applied a voom transformation to calculate precision weights for linear modeling. Next, we modeled the individual as a random effect using the duplicateCorrelation function and transformed the data with the correlation adjustment. Finally, we contrasted each treatment against the vehicle at each timepoint. Differentially enriched regions are defined as those regions for each treatment-vehicle pair that meet an adjusted *P* value threshold of < 0.05.

### Integrating DARs with genome annotations and other data sets

**Identifying gene feature enrichment in DARs and TOP2B-bound regions.** We analyzed the distribution of TOP2B-bound regions and DARs across gene features including promoter regions, exons, and introns using the annotation

package TxDb.Hsapiens.UCSC.hg38.knownGene (R package version 3.20.0). Gene feature enrichment was visualized using the plotAnnoBar function from the ChIPseeker package in R [60,61].

**Identifying TOP2B enrichment in DARs.** Each DAR and CAR treatment-time set was overlapped with TOP2B-bound regions. DAR vs CAR enrichment was determined by counting the number of unique regions that overlap a TOP2B region and computing odds ratios using a 2 x 2 contingency table. Significance was determined using a Fisher's exact test. DARs with $P < 0.05$ were determined to be enriched for TOP2B-bound regions.

**Identifying TE enrichment in DARs.** We obtained repeat annotations and genomic coordinates for hg38 from the RepeatMasker track [64] from the UCSC Table browser. We intersected the RepeatMasker track with our accessible chromatin regions in each DAR using the join_overlap_intersect function from the plyranges [56] package in R. We examined regions that overlapped the repeat regions by at least 1 bp. We stratified TEs by TE class: LINE, SINE, DNA, LTR, and SVA. Global TE or TE class enrichment was determined for each DAR set compared to the CAR set for each treatment time by computing odds ratios using a 2 x 2 contingency table, testing for significance using a Chi-square test and a Fisher exact test for counts $< 10$. Bonferroni adjustment for multiple testing was performed across categories of genome features. Regions with adjusted $P < 0.05$ were determined to be enriched for TEs. We repeated this analysis using nine representative TE families across classes as described above.

**Identifying CpG island enrichment in DARs.** We obtained hg38 CpG island annotations from the UCSC Table Browser [59] and overlapped these regions with DARs and CARs. We counted all regions that overlapped by at least 1 bp. Enrichment was determined by computing odds ratios using a 2 x 2 contingency table, testing for significance using a Chi-square test between each DAR set and CAR set at each treatment time. Bonferroni adjustment for multiple testing was performed across categories of genome features. Regions with adjusted $P < 0.05$ were determined to be enriched for CpG islands.

**Identifying TSS enrichment in DARs.** We obtained TSS annotations from TxDb.Hsapiens.UCSC.hg38.knownGene package (R package version 3.20.0) and counted our open chromatin regions that overlapped these TSS by at least 1 bp for each DAR and CAR set. The total number of overlapping regions in each DAR set was compared to the number in the CAR set. Enrichment was determined by computing odds ratios using a 2 x 2 contingency table, testing for significance using a Chi-square test followed by Bonferroni adjustment for multiple testing across categories of genome features. Sets of regions with adjusted $P < 0.05$ were determined to be enriched for TSS.

**Identifying heart regulatory element enrichment in DARs.** We obtained a list of candidate *cis*-Regulatory Elements (cREs) from the heart-left ventricle of a female adult (46 years) from the Search Candidate cis-Regulatory Elements by ENCODE (SCREEN) database (https://screen.wenglab.org/downloads on 04/13/24). cREs were further stratified into PLS, dELS, pELS, and CTCF. We overlapped the cREs (at least 1 bp) with our DARs and CARs within each treatment-time set. Enrichment was determined by computing odds ratios using a 2 x 2 contingency table, testing for significance using a Chi-square test followed by Bonferroni adjustment across categories of genome features. Regions with adjusted $P < 0.05$ were determined to be enriched for cREs.

**Integrating ATAC-seq data with H3K27ac CUT&Tag data.** 155,557 ATAC open chromatin regions were overlapped with 20,137 H3K27ac-enriched regions to find shared chromatin regions using the R package plyranges [56]. Shared regions are defined as regions which overlap by at least 1 bp. A total of 19,894 (98.8%) H3K27ac regions overlap open chromatin regions.

Response to drug treatment was compared across data types using the median $\log_2$ fold change across drug treatments at each timepoint.

**Integrating ATAC-seq data with CUT&Tag data and RNA-seq data.** Accessible chromatin regions were assigned a nearest gene using the closest TSS to the start of the region either upstream or downstream. This was made possible using the annotatePeak function from the ChIPseeker package in R and the TxDb.Hsapiens.UCSC.hg38.knownGene annotation package.

For integration with H3K27ac and RNA-seq data, H3K27ac enriched regions were overlapped with accessible regions and then filtered to include only those overlapping regions that were within 2 kb of an assigned TSS. These region-gene pairs were then filtered to only include those regions that were paired within 2 kb of an expressed gene from a matched RNA-seq data set (n = 14,084) [16]. This created a set of 10,859 region-expressed gene pairs used to calculate and plot the median log$_2$ fold change.

**Integrating DARs with differentially expressed genes.** Accessible regions were assigned to the nearest gene TSS and filtered to include those within 2 or 20 kb. The regions were then classified as DARs or CARs by treatment-time. The assigned nearest genes to each of these regions was also classified as either a differentially expressed gene (DEG) or a non-DEG using treatment-time specific DEGs as defined by Matthews *et al*. [16]. A 2 x 2 contingency matrix was constructed and the odds ratio calculated using a Fisher's exact test for DARs and CARs and DEGs and non-DEGs for each treatment-time set.

**Identifying TF motif enrichment in DEG-associated DARs.** We used the MEME Suite docker container (https://meme-suite.org v5.5.6) [65,66] with the xstreme tool [67] that performs comprehensive motif analysis on sequences which may contain motif sites anywhere throughout the given sequences. We first created DAR sets for each treatment-time by filtering out regions that were > 20 kb from the nearest TSS. We then filtered the list of DARs into DAR-DEGs and DAR-non-DEGs sets using the list of DEGs from our RNA-seq data of these cells [16]. DAR sets were centered and clipped to be 200 bp +/- the center of the region for a total length of 400 bp. We used the Jaspar 2022 core vertebrates non-redundant motif database to identify known motifs. The following settings were used with xstreme, --time 480 --streme-totallength 8,000,000 --meme-searchsize 100,000 --desc description --dna --evt 0.05 --minw 6 --maxw 15 --align center --meme-mod zoops --m motif_databases/JASPAR/JASPAR2022_CORE_vertebrates_non-redundant_v2.meme. Both the enrichment score and proportion of regions containing a given motif are reported.

**Pathway enrichment analysis of DAR-associated DEGs.** We identified the set of DEGs that have a DAR within 20 kb of the DEG TSS for each treatment time set. A unique gene list was gathered for each treatment-time set. KEGG pathway analysis was performed using the gost function from with the package gProfiler2 [68,69]. All expressed genes (n = 14,084) were used as a background. Significance was determined using FDR < 0.05.

**Integrating DARs with CVD SNPs.** AC-induced cardiotoxicity-associated SNPs: We obtained 108 SNPs associated with cardiotoxicity from a cohort of ~3,000 breast cancer patients treated with an AC [10]. The location of the SNP was obtained from the hg38 reference genome using Ensembl VEP (ensemble.org). To associate regions near SNPs, we downloaded a list of TAD domains from heart left ventricle from https://3dgenome.fsm.northwestern.edu/publications.html [24,25], and associated each SNP with a TAD and each accessible region with a TAD using the plyranges function join_overlap_intersect. The distance between each SNP and each region within a TAD was calculated using a custom analysis script from R. Significance was determined between all distances of DAR-SNP pairs within a TAD vs all CAR-SNP pairs within a TAD using the Wilcoxon test with *P* < 0.05 considered significant.

For all SNPs overlapping an open chromatin region, the rsID was entered into the GTEx portal (https://www.gtexportal.org/home/) to determine if the SNP is an eQTL for any gene in left ventricle heart tissue. The data were obtained on 01/30/2025.

Atrial fibrillation- and heart failure-associated SNPs: We obtained a list of SNPs for atrial fibrillation (AF), heart failure (HF), coronary artery disease (CAD), and ischemic heart disease (IHD) from the GWAS Catalog (https://www.ebi.ac.uk/gwas/home) [12]. AF has 679 unique associated SNPs, HF has 403 unique associated SNPs, CAD has 2,320 unique associated SNPs, and IHD seven unique associated SNPs. Given the large number of SNPs we considered only those SNPs that directly overlap with accessible chromatin regions. Enrichment testing of SNPs in DAR vs CAR treatment-time sets was performed using odds ratios from a 2 x 2 contingency matrix. We tested for significance using a Chi-square test or a Fisher's exact test for counts < 10. *P* < 0.05 was determined to be significant.

All custom analysis scripts are available on github through workflowr [70].

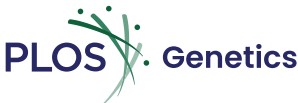

## Supporting information

**S1 Table. ATAC-seq sample metadata.**
(XLSX)

**S2 Table. Gamma H2A.X quantification.**
(XLSX)

**S3 Table. ATAC-seq read number and peak number by sample.**
(XLSX)

**S4 Table: Pairwise differential accessibility analysis for DOX vs VEH at 3 hours.**
(XLSX)

**S5 Table. Pairwise differential accessibility analysis for DNR vs VEH at 3 hours.**
(XLSX)

**S6 Table. Pairwise differential accessibility analysis for EPI vs VEH at 3 hours.**
(XLSX)

**S7 Table. Pairwise differential accessibility analysis for MTX vs VEH at 3 hours.**
(XLSX)

**S8 Table. Pairwise differential accessibility analysis for TRZ vs VEH at 3 hours.**
(XLSX)

**S9 Table. Pairwise differential accessibility analysis for DOX vs VEH at 24 hours.**
(XLSX)

**S10 Table. Pairwise differential accessibility analysis for DNR vs VEH at 24 hours.**
(XLSX)

**S11 Table. Pairwise differential accessibility analysis for EPI vs VEH at 24 hours.**
(XLSX)

**S12 Table. Pairwise differential accessibility analysis for MTX vs VEH at 24 hours.**
(XLSX)

**S13 Table. Pairwise differential accessibility analysis for TRZ vs VEH at 24 hours.**
(XLSX)

**S14 Table. ATAC-seq open chromatin regions and associated characteristics.**
(XLSX)

**S15 Table. H3K27ac CUT&Tag sample metadata.**
(XLSX)

**S16 Table. H3K27ac CUT&Tag read and peak number by sample.**
(XLSX)

**S17 Table. Pairwise differential acetylation analysis for DOX vs VEH at 3 hours.**
(XLSX)

**S18 Table. Pairwise differential acetylation analysis for DNR vs VEH at 3 hours.**
(XLSX)

**S19 Table. Pairwise differential acetylation analysis for EPI vs VEH at 3 hours.**
(XLSX)

**S20 Table. Pairwise differential acetylation analysis for MTX vs VEH at 3 hours.**
(XLSX)

**S21 Table. Pairwise differential acetylation analysis for DOX vs VEH at 24 hours.**
(XLSX)

**S22 Table. Pairwise differential acetylation analysis for DNR vs VEH at 24 hours.**
(XLSX)

**S23 Table. Pairwise differential acetylation analysis for EPI vs VEH at 24 hours.**
(XLSX)

**S24 Table. Pairwise differential acetylation analysis for MTX vs VEH at 24 hours.**
(XLSX)

**S1 Fig. Cardiac differentiation generates high purity cardiomyocytes. (A)** Flow cytometry results indicating the proportion of TNNT2-positive cells following differentiation in Individuals A, B, C, D as well as unlabeled iPSC-CMs from a mixture of Individuals A and D. **(B)** Proportion of TNNT2-positive cells from each individual (A: orange; B: purple; C: magenta; D: teal). Data from each individual is presented as the mean of two replicate flow cytometry experiments.
(TIF)

**S2 Fig. Read numbers are similar across time and drug treatments.** The number of total reads, total mapped reads, total nuclear-mapped reads, unique-nuclear mapped reads, and deduplicated-unique-nuclear mapped reads by time and treatment for four individuals. Samples for each individual (A: orange; B: purple; C: magenta; D: teal) are grouped by treatment and time. Read numbers represent the sum of read 1 and read 2 for each sample.
(TIF)

**S3 Fig. Open chromatin region numbers are similar across time and drug treatments. (A)** Number of unique-deduplicated reads (read 1 + read 2 per sample) across time and treatment (DOX: mauve; EPI: pink; DNR: yellow; MTX: blue; TRZ: olive; VEH: green) at three and 24 hours for each individual (A: orange; B: purple; C: magenta; D: teal). **(B)** Number of open chromatin regions called per treatment and time for each individual.
(TIF)

**S4 Fig. Samples have a high fraction of read-fragments in high-confidence open chromatin regions.** The fraction of fragments in a set of high-confidence open chromatin regions (n = 172,481) for each individual (A: orange; B: purple; C: magenta; D: teal) across time and treatment (DOX: mauve; EPI: pink; DNR: yellow; MTX: blue; TRZ: olive; VEH: green). All samples meet or exceed the minimum acceptable quality metric of 0.2 used by ENCODE (dashed red line).
(TIF)

**S5 Fig. iPSC-CM open chromatin regions are shared with human heart left ventricle open chromatin regions. (A)** Overlap between our 155,557 filtered high-confidence open chromatin regions in iPSC-CMs and open chromatin regions from heart left ventricle tissue from a 41-year-old female (ATAC-seq file ENCF966JZT from ENCODE experiment ENCSR204PZT). **(B)** Permutation test to determine significance of the overlap.
(TIF)

**S6 Fig. Open chromatin regions are enriched at transcription start sites. (A)** Count frequency of fragments that map within +/-1.5 kb of all transcription start sites (TSS) across samples by individual and time. The dashed line is the TSS

location. Solid lines are colored by treatment (DOX: mauve; EPI: pink; DNR: yellow; MTX: blue; TRZ: olive; VEH: green). **(B)** TSS enrichment scores across individuals by treatment and time. Dots represent treatment for the individuals listed above. The ENCODE ATAC-seq TSS enrichment score thresholds for the minimum ideal (solid blue line) and minimum acceptable threshold (dotted black line) are also shown.
(TIF)

**S7 Fig. Genome coverage is similar across samples at the TSS of the cardiac gene *TNNT2*.** ATAC-seq fragments at a representative open chromatin region (chr1:201,376,390–201,378,217) at the TSS of troponin T (*TNNT2*) for all samples across time and drug treatments. Three-hour samples are shown on the left, 24-hour samples are on the right. Samples are grouped by individual and colored by treatment (DOX: mauve; EPI: pink; DNR: yellow; MTX: blue; TRZ: olive; VEH: green).
(TIF)

**S8 Fig. ATAC-seq samples cluster by time and treatment.** Pearson correlation of $\log_2$ cpm values across all pairs of samples for the high-confidence set of 155,557 open chromatin regions. Top bars are colored by individual (A: orange; B: purple; C: magenta; D: teal), treatment (DOX: mauve; EPI: pink; DNR: yellow; MTX: blue; TRZ: olive; VEH: green), time (three hours: pink; 24 hours: brown), class (anthracycline: yellow; non-anthracycline: orange), and classification as a TOP2i (no: light green; yes: dark green).
(TIF)

**S9 Fig. PC1 associates with drug treatment and PC2 associates with individual.** Demonstration of variance contributed to the first two principal components (PC) by individual, treatment, and time (the three major biological factors in this study). **(A)** Variance of individual (A: orange; B: purple, C: magenta; D: teal) as a function of PC1 and PC2. The correlation between individual and each PC is calculated using a linear model. *P* values represent the significance of the F-statistic from the model. **(B)** Variance of treatment (DOX: mauve; EPI: pink; DNR: yellow; MTX: blue; TRZ: olive; VEH: green) as a function of PC1 and PC2. **(C)** Variance of time (three hour: pink; 24 hour: brown) as a function of PC1 and PC2.
(TIF)

**S10 Fig. Thousands of chromatin regions show changes in accessibility in response to TOP2i treatment.** Volcano plots representing open chromatin regions that are differentially accessible (adjusted *P*<0.05) in each drug treatment compared to VEH at each timepoint. Regions that increase in accessibility in response to treatment (opening) are represented in blue, and regions that significantly decrease in accessibility (closing) are represented in red.
(TIF)

**S11 Fig. Drug treatment and VEH show distinct chromatin accessibility at DARs.** Chromatin accessibility ($\log_2$ cpm) at regions classified as DARs across treatments. Given that there is only one DAR in response to TRZ treatment, all accessible regions (ARs) are represented for this treatment. A common color scale was applied to all heatmaps to allow consistent comparisons across conditions. The scale was based on the global distribution of $\log_2$ cpm values across all samples. To minimize the influence of outlier expression values, the color scale was capped at the first and 99th percentiles of the global $\log_2$ cpm distribution, mapping the lowest 1% to blue and the highest 1% to red. The median $\log_2$ cpm value was set to white.
(TIF)

**S12 Fig. Top differentially accessible regions are shared across anthracyclines. (A)** Chromatin accessibility ($\log_2$ cpm) at top differentially accessible regions for DOX, EPI, and DNR at three and 24 hours across each treatment (DOX: mauve; EPI: pink; DNR: yellow; MTX: blue; TRZ: olive; VEH: green). Asterisk represents regions classified as a DAR following each treatment (adjusted *P*<0.05).
(TIF)



**S13 Fig. The MIR SINE family is enriched amongst AC DARs at both timepoints.** Enrichment of TE families in DARs compared to CARs for each drug treatment determined by Chi-square test and Benjamini-Hochberg multiple testing correction. Asterisk represents TE families that are significantly enriched.
(TIF)

**S14 Fig. H3K27ac regions are enriched at transcription start sites.** Count frequency of H3K27ac regions that map within +/-1.5 kb of all TSS in the hg38 genome across samples by individual and time. The dashed line is the TSS location. Solid lines are colored by treatment (DOX: mauve; EPI: pink; DNR: yellow; MTX: blue; VEH: light green). Individual plots with fewer than five solid lines indicate that those library preparations failed.
(TIF)

**S15 Fig. H3K27ac CUT&Tag data separate by time and drug treatment.** Pearson correlation of $\log_2$ cpm values across 20,137 high-confidence H3K27ac-enriched regions. Colored bars represent individual (A: orange; B: purple; C: magenta; D: teal), treatment (DOX: mauve; EPI: pink; DNR: yellow; MTX: blue; TRZ: olive; VEH: green), time (three hours: pink; 24 hours: brown), class (anthracycline: yellow; non-anthracycline: orange), and classification as a TOP2i (no: light green; yes: dark green).
(TIF)

**S16 Fig. Change in H3K27ac enrichment in response to drugs is highly correlated across ACs. (A)** Volcano plots representing H3K27ac enrichment changes for each drug compared to VEH at each timepoint across treatments (adjusted $P < 0.05$). Regions that increase in acetylation in response to treatment are represented in blue (up), and regions that decrease in acetylation in response to treatment are represented in red (down). **(B)** Pearson correlation of H3K27ac drug response across time and treatment. Colored bars represent time (three hours: pink; 24 hours: brown), and class (anthracycline: yellow; non-anthracycline: orange).
(TIF)

**S17 Fig. Shared ATAC and H3K27ac region response to TOP2i clusters by time.** Pearson correlation of $\log_2$ fold change of drug response in regions shared between ATAC-seq and H3K27ac CUT&Tag data. Shared regions are defined as regions which overlap by at least 1 bp. A total of 19,894 (98.8%) H3K27ac regions overlap open chromatin regions. Top bars are colored by molecular phenotype (ATAC-seq: dark blue; H3K27ac CUT&Tag: maroon), treatment (DOX: mauve; EPI: pink; DNR: yellow; MTX: blue), time (three hours: pink; 24 hours: brown), and class (anthracycline: yellow; non-anthracycline: orange).
(TIF)

**S18 Fig. Shared accessible and H3K27ac chromatin regions have a significant correlation with nearby gene expression at 24 hours.** Correlation between drug response of open chromatin regions shared with H3K27ac regions and drug response of nearby genes, and correlation between drug response of open chromatin regions that do not overlap H3K27ac regions and drug response of nearby genes. Only open chromatin regions that are within +/- 2 kb of an expressed gene TSS [16] are shown (n = 10, 859).
(TIF)

**S19 Fig. Drug response associates with time and molecular phenotype.** 10,859 open chromatin regions with H3K27ac enrichment were associated with the nearest expressed gene within 2 kb of the TSS [16]. A Pearson correlation of the $\log_2$ fold change in response to each drug was performed for three molecular phenotypes (gene expression, chromatin accessibility and H3K27ac enrichment). Top bars are colored by molecular phenotype (RNA-seq: purple; ATAC-seq: dark blue; H3K27ac CUT&Tag: maroon), treatment (DOX: mauve; EPI: pink; DNR: yellow; MTX: blue), and time (three hours: pink; 24 hours: brown).
(TIF)

**S20 Fig. DARs are present within anthracycline-induced cardiotoxicity SNP-containing TADs.** Drug response effect size ($\log_2$ fold change) of accessible regions within the same TAD as cardiotoxicity SNPs. Asterisk represents regions that are classified as DARs in each treatment.
(TIF)

**S21 Fig. DARs overlap AF SNPs.** Drug response effect size ($\log_2$ fold change) of accessible regions overlapping AF SNPs. Asterisk represents regions that are classified as DARs in each treatment.
(TIF)

**S22 Fig. DARs overlap HF SNPs.** Drug response effect size ($\log_2$ fold change) of accessible regions overlapping HF SNPs. Asterisk represents regions that are classified as DARs in each treatment.
(TIF)

## Acknowledgments

We thank all members of the Ward Lab, especially Omar Johnson, for helpful discussions. We thank Kelly Frazer and the University of California San Diego for providing the iPSC lines through the iPSCORE resource. We thank the Next Generation Sequencing Core Facility at the University of Texas Medical Branch for sequencing the ATAC-seq and CUT&Tag libraries. The Genotype-Tissue Expression (GTEx) Project was supported by the Common Fund of the Office of the Director of the National Institutes of Health, and by NCI, NHGRI, NHLBI, NIDA, NIMH, and NINDS. The data used for the analyses described in this manuscript were obtained from the GTEx Portal on 01/30/25. The authors acknowledge the Texas Advanced Computing Center (TACC) at The University of Texas at Austin for providing HPC resources that have contributed to the research results reported within this paper (http://www.tacc.utexas.edu).

## Author contributions

**Conceptualization:** Michelle C. Ward.

**Formal analysis:** E. Renee Matthews, Raodatullah O Abodunrin, Sayan Paul, Michelle C. Ward.

**Funding acquisition:** Michelle C. Ward.

**Investigation:** E. Renee Matthews, Raodatullah O Abodunrin, John D Hurley, Sayan Paul, José A Gutierrez, Alyssa R Bogar.

**Supervision:** Michelle C. Ward.

**Writing – original draft:** E. Renee Matthews, Michelle C. Ward.

**Writing – review & editing:** John D Hurley, Sayan Paul, José A Gutierrez, Alyssa R Bogar.

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
