## [Decision Letter · Decision Letter 0]

3 Jun 2025

PGENETICS-D-25-00511

Anthracyclines induce global changes in cardiomyocyte chromatin accessibility that overlap with cardiovascular disease loci

PLOS Genetics

Dear Dr. Ward,

Thank you for submitting your manuscript to PLOS Genetics. After careful consideration, we feel that it has merit but does not fully meet PLOS Genetics's publication criteria as it currently stands. Therefore, we invite you to submit a revised version of the manuscript that addresses the points raised during the review process.

Please submit your revised manuscript within 60 days Aug 02 2025 11:59PM. If you will need more time than this to complete your revisions, please reply to this message or contact the journal office at plosgenetics@plos.org. Please include the following items when submitting your revised manuscript:

We look forward to receiving your revised manuscript.

Kind regards,

Anthony B Firulli

Academic Editor

PLOS Genetics

John Greally

Section Editor

PLOS Genetics

Aimée Dudley

Editor-in-Chief

PLOS Genetics

Anne Goriely

Editor-in-Chief

PLOS Genetics

**Journal Requirements:**

3) Please ensure that the funders and grant numbers match between the Financial Disclosure field and the Funding Information tab in your submission form. Note that the funders must be provided in the same order in both places as well.

**Reviewers' comments:**

Reviewer's Responses to Questions

**Comments to the Authors:**

Reviewer #1: The Article entitled “Anthracyclines induce global changes in cardiomyocyte chromatin accessibility that overlap with cardiovascular disease loci” is an investigation of chromatin accessibility profile of human induced pluripotent stem cell derived cardiomyocytes treated with anthracyclines for 3 and 24 hrs. The authors generated chromatin accessibility profile by performing ATACseq and conducted analysis of the data to classify a set of regions that open/close early, remain open/close till 24hrs or not respond to anthracycline drugs. Further, the authors found that these regions are bound by DNA damage specific transcription factors and occupy with some cardiovascular disease risk specific loci. Overall, this is a descriptive study that provides little insights into the mechanisms of why anthracyclines cause heart disease including heart failure and arrhythmias.

Major comments:

1. While breast cancer patients are treated for long time with Anthracyclines, the current study evaluates their effect on iPSCs derived heart cells for 3 and 24 hrs. It is not clear why was this time points chosen. Even with 3 and 24hrs, there is stark differences in chromatin accessibility profiles, indicating a different treatment period will have more chromatin changes, thus it is difficult to infer any valid conclusions from these datasets.

2. Since anthracyclines only affect myocardial function including causing heart failure and arrythmias and do not affect ischemic heart diseases, can the authors do similar experiments in endothelial cells and show that they do not affect endothelial cell chromatin accessibility?

3. Mature human cardiomyocytes are bi-nucleated. It appears that the authors used iPSC derived immature cardiomyocytes. Please provide data to show that the cardiomyocytes used here are mature cardiomyocytes that mimics patient conditions.

4. Does anthracycline treated human iPSC cardiomyocytes show arrhythmia and heart failure phenotype? In absence of these phenotypic observations, chromatin accessibility changes, although informative, do not contribute much insight into drug induced cardiotoxicity. The authors should provide electrophysiology data of the iPSC cardiomyocytes after anthracycline treatment to justify the timing and concentration of the drugs used.

5. In absence of HiC data, associating accessibility regions and cardiovascular diseases risk loci within 20 kb is bit of a stretch. Can the authors perform HiC to prove that these accessibility regions and the CVD risk loci are in the same loop or TAD.

6. Overall, the molecular basis of anthracycline induced gene expression and cardiotoxicity is not clear from this study.

Minor:

1. Line 33: it appears that iPSC cardiomyocytes were derived from four individuals treated with anthracyclines: please revise to reflect that iPSC cardiomyocytes derived from four healthy individuals were treated with anthracyclines to evaluate their role on chromatin accessibilities.

2. Line 167: left ventricle?

3. Line 201: Is it log10 fold change or log2 fold change?

Reviewer #2: How breast cancer drugs cause cardiovascular disease, such as heart failure or atrial fibrillation, is unclear. Here, Matthews et al. analyzed bulk chromatin accessibility of iPSC-derived cardiomyocytes from four females treated with 5 breast cancer drugs that induce cardiac dysfunction or a vehicle control for 3 or 24 hours. The authors observed patterns of responses in chromatin accessibility, which was analogous to their previous work describing patterns for gene expression. The authors identified putative transcription factor motifs, including those TFs associated with DNA damage, in differentially accessible regions upon treatment. The authors observed that chromatin accessibility overlapped with histone modifications such as H3K27Ac. Further, GWAS variants from breast cancer patients who developed cardiac dysfunction upon treatment with these medications, or variants for heart failure or atrial fibrillation, overlapped with these regions of differentially accessible chromatin, suggesting genetic and environmental contributions for cardiovascular risk. In general, the authors have generated data from 48 conditions, and the manuscript is well written. Data from Fig 7 seems to be most impactful if SNPs from heart failure or atrial fibrillation are statistically enriched in differentially accessible regions by cancer therapies in iPSC-derived cardiomyocytes. However, beyond this, the work in its current form provides incremental biological insights.

Some comments are below.

Major:

1. I am somewhat confused by how the data is presented. Some of the data is presented in aggregate as a box plot by condition, which shows the bulk variation among individuals. This conveys the message that in general, individuals respond in a similar way, and that the message is that most of the treatments cause a similar pattern of responses at the level of accessible chromatin. I believe that this may be the intended message. However, selected browser tracks show each of the conditions per individual, showing the response to treatment in each individual. This presentation conveys that a given individual may respond differently at the level of chromatin accessibility to the various treatments. Unfortunately, this mix of presentation occurs throughout the manuscript.

2. Further, the data is presented in aggregate based on time point and treatment. Many correlation plots are presented for all of the conditions. However, I don’t have a good sense about how similar the samples are within a group (ie. vehicle by individuals A-D) at the level individual accessible chromatin regions, except for some selected browser tracks. I think that this could be accomplished by a heatmap of statistically-robust differentially accessible peaks. (Apologies if it’s presented, and I have missed it). This information would be important to convey the variation within each condition by individual (ie. showing Vehicle for individuals A-D) for then comparing with treatment conditions (ie. Treatment X for individuals A-D). If I understand correctly, this would align with how the data was analyzed for treatment-vehicle comparisons, although this is obscured in how the data is currently presented.

3. I can appreciate the effort to stratify the accessibility responses using the joint Bayesian into shared patterns. However, it is not clear to me if relevant biological insights have been derived from this.

4. I understand that many of the acronyms correspond to those in the authors’ previous 2024 publication. However, the number of acronyms, especially in Figure 1, borders on overwhelming, especially as many acronyms, if not most, are not common.

5. Although a lot of data is presented, it’s not clear to me the relevance of Fig 1-4, as presented.

6. I see p-values for some of the data in the manuscript, but it is not apparent to me for others. (Apologies if I have missed this). Please also consider adding p-values in the Figures or the figure legends. For example, For Fig S4, is the overlap with the in vivo heart sample statistically significant? For Fig 6 and 7, are SNPs significantly enriched in differentially accessible chromatin regions? Same with the comparisons in B, C, and D for Fig 6 or C, D, and E for Fig 7. Are SNPs from heart failure or atrial fibrillation are statistically enriched in differentially accessible regions by cancer therapies in iPSC-derived cardiomyocytes? If yes, please show the relevant adj p-value by statistical metric.

Minor:

1. I assume the authors refer “heart eSNP” as a SNP in a heart enhancer, although I could be mistaken. This ambiguity could be clarified for the reader with a definition and a reference to the publication that defines this set of eSNPs.

**Have all data underlying the figures and results presented in the manuscript been provided?**

Reviewer #1: Yes

Reviewer #2: Yes

PLOS authors have the option to publish the peer review history of their article (what does this mean? ). If published, this will include your full peer review and any attached files.

**Do you want your identity to be public for this peer review?** For information about this choice, including consent withdrawal, please see our Privacy Policy .

Reviewer #1: No

Reviewer #2: No

**Figure resubmission:**
---

## [Decision Letter · Decision Letter 1]

4 Sep 2025

PGENETICS-D-25-00511R1

Anthracyclines induce global changes in cardiomyocyte chromatin accessibility that overlap with cardiovascular disease loci

PLOS Genetics

Dear Dr. Ward,

Thank you for submitting your manuscript to PLOS Genetics. After careful consideration, we feel that it has merit but does not fully meet PLOS Genetics's publication criteria as it currently stands. Therefore, we invite you to submit a revised version of the manuscript that addresses the points raised during the review process.

Please submit your revised manuscript within 30 days Oct 04 2025 11:59PM. If you will need more time than this to complete your revisions, please reply to this message or contact the journal office at plosgenetics@plos.org. Please include the following items when submitting your revised manuscript:

We look forward to receiving your revised manuscript.

Kind regards,

Anthony B Firulli

Academic Editor

PLOS Genetics

John Greally

Section Editor

PLOS Genetics

Aimée Dudley

Editor-in-Chief

PLOS Genetics

Anne Goriely

Editor-in-Chief

PLOS Genetics

**Reviewers' comments:**

Reviewer's Responses to Questions

Reviewer #1: The revised manuscript “Anthracyclines induce global changes in cardiomyocyte chromatin accessibility that overlap with cardiovascular disease loci” has addressed many of my previous comments and added new data. The discovery of the gained and lost chromatin accessibility regions due to anthracycline treatment on human iPSC derived cardiomyocytes and their proximity to atrial fibrillation and heart failure genetic risk sites provides insights into why these drugs are associated atrial fibrillation and heart failure in patients. What is does not prove is a mechanistic link of how and why chromatin regions are remodeled near these disease risk loci. Also, it does not provide any new insights into when and where anthracyclines should or should not be used in breast cancer treatment and when the risk of cardiotoxicity and cardiovascular disease risk outweighs treatment benefits. This article provides simple association of disease risk SNPs and treatment induced chromatin accessibility changes, which should be of broad interest, however, it doesn’t go beyond this to provide applications or implications of this study, which the authors should discuss more.

Minor points

1. Fig1b: authors should high quality images of show single cardiomyocytes

2. Table S1 shows >98% cardiomyocytes, but authors should show a FACS plot showing the purity of cardiomyocytes

3. Fig 3a, no input sequence shown or peak called shown making it difficult to understand if the ChIPseq signal is real or background

4. Line 244: Fig 3A, should be Fig 3F

Reviewer #2: I think that Matthews et al.’s additions to the re-focused manuscript have made it much improved. The authors have addressed many of my concerns. A few minor comments are below.

Minor comments:

1) I think that the comparison of DARs to CARs is useful in many contexts of the manuscript, especially when showing significance in DARs but not CARs. However, I’m not aware of other publications that supports the notion that the proximity of a SNP to DARs compared to CARs is meaningful, especially when chromatin is often considered in a 3-D context. Please include support for this if available in the literature. Otherwise, I think that the few mentions of the comparison of the distance of SNP to DARs and CARs (and the accompanying figure panel) of this could be amended without substantially weakening the manuscript.

2) I understood that SNPs of ischemic heart disease and coronary artery disease are used to contrast to SNPs of heart failure and atrial fibrillation. If this is the intention, then the way that line 690 is written makes this point a little confusing.

3) In the paragraph for lines 786-801, I was a little unclear what sentences related to the current work versus discussion of previous publications. For example, line 793-794 is not clear to me what the authors are trying to say.

4) Line 877: unless I missed it, I did not see occupancy data for JUN/FOS or p53. I would suggest slight modification of “binding” to something like “accessibility of motifs”…

**Have all data underlying the figures and results presented in the manuscript been provided?**

Reviewer #1: Yes

Reviewer #2: Yes

PLOS authors have the option to publish the peer review history of their article (what does this mean? ). If published, this will include your full peer review and any attached files.

**Do you want your identity to be public for this peer review?** For information about this choice, including consent withdrawal, please see our Privacy Policy .

Reviewer #1: No

Reviewer #2: **Yes: ** Irfan S. Kathiriya

**Figure resubmission:**
---

## [Decision Letter · Decision Letter 2]

30 Sep 2025

Dear Dr Ward,

We are pleased to inform you that your manuscript entitled "Anthracyclines induce global changes in cardiomyocyte chromatin accessibility that overlap with cardiovascular disease loci" has been editorially accepted for publication in PLOS Genetics. Congratulations!

Yours sincerely,

Anthony B Firulli

Academic Editor

PLOS Genetics

John Greally

Section Editor

PLOS Genetics

Aimée Dudley

Editor-in-Chief

PLOS Genetics

Anne Goriely

Editor-in-Chief

PLOS Genetics

BlueSky: @plos.bsky.social

Comments from the reviewers (if applicable):

Reviewer's Responses to Questions

**Comments to the Authors:**

Reviewer #1: The authors have addressed most of my comments and explained better in the revised manuscript.

Reviewer #2: In my view, the authors have sufficiently addressed the reviewers' comments. I have no additional comments.

**Have all data underlying the figures and results presented in the manuscript been provided?**

Reviewer #1: Yes

Reviewer #2: Yes

PLOS authors have the option to publish the peer review history of their article (what does this mean? ). If published, this will include your full peer review and any attached files.

**Do you want your identity to be public for this peer review?** For information about this choice, including consent withdrawal, please see our Privacy Policy .

Reviewer #1: No

Reviewer #2: **Yes: ** Irfan S. Kathiriya

**Data Deposition**

http://datadryad.org/submit?journalID=pgenetics&manu=PGENETICS-D-25-00511R2

**Press Queries**

---

## [Editor Report · Acceptance letter]

PGENETICS-D-25-00511R2

Anthracyclines induce global changes in cardiomyocyte chromatin accessibility that overlap with cardiovascular disease loci

Dear Dr Ward,

We are pleased to inform you that your manuscript entitled "Anthracyclines induce global changes in cardiomyocyte chromatin accessibility that overlap with cardiovascular disease loci" has been formally accepted for publication in PLOS Genetics! Your manuscript is now with our production department and you will be notified of the publication date in due course.

With kind regards,

Anita Estes

PLOS Genetics

On behalf of:
